# Bayes Conditional Distribution Estimation for Knowledge Distillation Based on Conditional Mutual Information

**Linfeng Ye**\*, **Shayan Mohajer Hamidi**\*, **Renhao Tan**\* & **En-Hui Yang**
Department of Electrical and Computer Engineering, University of Waterloo
{l44ye,smohajer,cameron.tan,ehyang}@uwaterloo.ca

## Abstract

It is believed that in knowledge distillation (KD), the role of the teacher is to provide an estimate for the unknown Bayes conditional probability distribution (BCPD) to be used in the student training process. Conventionally, this estimate is obtained by training the teacher using maximum log-likelihood (MLL) method. To improve this estimate for KD, in this paper we introduce the concept of conditional mutual information (CMI) into the estimation of BCPD and propose a novel estimator called the maximum CMI (MCMI) method. Specifically, in MCMI estimation, both the log-likelihood and CMI of the teacher are simultaneously maximized when the teacher is trained. Through Eigen-CAM, it is further shown that maximizing the teacher's CMI value allows the teacher to capture more contextual information in an image cluster. Via conducting a thorough set of experiments, we show that by employing a teacher trained via MCMI estimation rather than one trained via MLL estimation in various state-of-the-art KD frameworks, the student's classification accuracy consistently increases, with the gain of up to 3.32%. This suggests that the teacher's BCPD estimate provided by MCMI method is more accurate than that provided by MLL method. In addition, we show that such improvements in the student's accuracy are more drastic in zero-shot and few-shot settings. Notably, the student's accuracy increases with the gain of up to 5.72% when 5% of the training samples are available to the student (few-shot), and increases from 0% to as high as 84% for an omitted class (zero-shot). The code is available at https://github.com/iclr2024mcmi/ICLRMCMI.

## 1 Introduction

Knowledge distillation (Buciluǎ et al., 2006; Hinton et al., 2015) (KD) has received tremendous attention from both academia and industry in recent years as a highly effective model compression technique, and has been deployed in different settings (Radosavovic et al., 2018; Furlanello et al., 2018; Xie et al., 2020). The crux of KD is to distill the knowledge of a cumbersome model (teacher) into a lightweight model (student). Followed by Hinton et al. (2015), many researchers tried to improve the performance of KD (Romero et al., 2014; Anil et al., 2018; Park et al., 2019b), and to understand why distillation works (Phuong & Lampert, 2019; Mobahi et al., 2020; Allen-Zhu & Li, 2020; Menon et al., 2021; Dao et al., 2020).

One critical component of KD that has received relatively little attention is the training of the teacher model. In fact, in most of the existing KD methods, the teacher is trained to maximize its own performance, even though this does not necessarily lead to an improvement in the student's performance (Cho & Hariharan, 2019; Mirzadeh et al., 2020). Therefore, to effectively train a student to achieve high performance, it is essential to train the teacher accordingly.

Recently, Menon et al. (2021) observed that the teacher's soft prediction can serve as a proxy for the true Bayes conditional probability distribution (BCPD) of label $y$ given an input $x$, and that the closer the teacher's prediction to BCPD, the lower the variance of the student objective function. On the other hand, (Dao et al., 2020) showed that the accuracy of the student is directly bounded

---

\*Authors contributed equally.

by the distance between teacher's prediction and the true BCPD through the Rademacher analysis. In addition, Yang et al. (2023) showed that the error rate of a deep neural network (DNN) is upper bounded by the average of the cross entropy of the BCPD and the DNN's output. This together with the observations in Menon et al. (2021); Dao et al. (2020) implies that the better the estimation of the BCPD by the teacher, the better the student's performance would be.

In conventional KD, the BCPD estimate is obtained by training the teacher using maximum log-likelihood (MLL) method. In this paper, we instead explore a different way to train the teacher for the purpose of providing a better BCPD estimate so as to improve the performance of KD. To this end, we introduce a novel BCPD estimator, the rationale behind which will be briefly explained in the sequel.

In a multi-class classification task, each of the "ground truth labels" can be regarded as a distinct *prototype*. In this sense, the images with the same label are different manifestations of a common *prototype*, each with its own context. For instance, consider the following two images whose *prototypes* are "dog": (i) a dog in a jungle, and (ii) a dog in a desert. The first and second images have some contexts related to jungle and desert, respectively, although both have the same *prototype*. As such, each manifestation (for instance, a training/testing image) contains two types of information: (i) its *prototype*; and (ii) some contextual information on top its *prototype*. Therefore, in KD, it would be desirable for the teacher, providing an estimate of BCPD, to be capable of providing good amount of information about both of these types of information for the input images.

A teacher trained using MLL solely aims at estimating the *prototypes*[1]. In order for the teacher to provide a good amount of the contextual information, first we have to find out how this information could be quantified. To this aim, we introduce an information quantity from information theory (Cover, 1999) to measure such information. Specifically, by treating the teacher model as a mapping function, we argue that this contextual information resides in conditional mutual information (CMI) $I(X; \hat{Y} \mid Y)$, where $X, Y$ and $\hat{Y}$ are three random variables representing the input image, the ground truth label, and the label predicted by the teacher, respectively. As such, $I(X; \hat{Y} \mid Y)$ quantifies how much information $\hat{Y}$ can tell about $X$, given the *prototype* $Y$; this aligns with the concept of contextual information. To provide a good amount of contextual information, a logical step would be to maximize the CMI value under certain constraints during the teacher's training process.

Henceforth, to balance the prototype and the contextual information, we train the teacher to simultaneously **maximize** (i) the log-likelihood (LL) of the *prototype* (which is equivalent to minimize its cross entropy loss), and (ii) its CMI value. We refer to such an estimator as maximum CMI (MCMI) method. As seen in section 6, although the classification accuracy of the teachers trained via MCMI is lower than that of the teachers trained via MLL, the former can consistently increase the accuracy of the student models. In addition, we show that MCMI is general in that it could be deployed in different knowledge transfer methods. In summary, the contributions of the paper are as follows:

• We argue that the so-called dark knowledge passed by the teacher to the student is indeed the contextual information of the images which could be quantified via teacher's CMI value.

• Equipped with the concept of CMI, we attribute the reason of using temperature in KD to blindly increase the teacher's CMI value.

• We develop a novel BCPD estimator dubbed MCMI that simultaneously maximizes the LL and CMI of the teacher. This enables the teacher to provide more accurate estimate of the true BCPD to the student, thereby enhancing the student's performance.

• We show that since larger models generally have lower CMI values, they are not often good teachers for KD. However, this issue is **mitigated** when teachers are trained using MCMI. We further explain why the models trained via early stopping are better teachers than those that are fully-trained.

• To demonstrate the effectiveness of MCMI, we conduct a thorough set of experiments over CIFAR-100 (Krizhevsky et al., 2012) and ImageNet (Deng et al., 2009) datasets, and show that by replacing the teacher trained by MLL with that trained by MCMI in the existing state-of-the-art KD methods, the student's accuracy consistently increases. Moreover, these improvements are even more significant in both few-shot and zero-shot settings.

---

[1] In this paper, we use the terms *prototype* and ground truth label interchangeably.

## 2 RELATED WORKS

• **Distillation losses in KD:** The concept of knowledge transfer, as a means of compression, was first introduced by Buciluă et al. (2006). Then, Hinton et al. (2015) popularized this concept by softening the teacher's and student's logits using temperature technique where the student mimics the soft probabilities of the teacher, and referred to it as KD. To improve the effectiveness of distillation, various forms of knowledge transfer methods have been introduced which could be mainly categorized into three types: (i) logit-based (Zhu et al., 2018; Stanton et al., 2021; Chen et al., 2020a; Li et al., 2020; Beyer et al., 2022; Zhao et al., 2022), (ii) representation-based (Romero et al., 2014; Zagoruyko & Komodakis, 2016; Yim et al., 2017; Chen et al., 2021b; Yang et al., 2021), and (iii) relationship-based (Park et al., 2019b; Liu et al., 2019; Peng et al., 2019; Yang et al., 2022) methods. Closely related to the scope of this paper, some methods in the literature deploy the concept of information theory in KD to make the distillation process more effective. We defer discussing these works to appendix A.1, and note that our method is different from these works in that we use information-theoretic concepts for training the teacher only, rather than for building strong relationship between the teacher and student.

• **Training a customized teacher:** In the literature, only a few works trained teachers specifically tailored for KD. Yang et al. (2019) attempted to train a tolerant teacher which provides more secondary information to the student. They realized this via adding an extra term to facilitating a few secondary classes to emerge to complement the primary class. Cho & Hariharan (2019); Wang et al. (2022) regularized the teacher utilizing early-stopping during the training. Tan & Liu (2022) trained the teachers to have more dispersed soft probabilities. Additionally, Dong et al. (2023) stated that a model exhibiting local Lipschitz continuity around training inputs can serves as a student-oriented teacher. Yet, our work distinguishes itself from these prior studies as it aims to estimate the true BCPD.

## 3 NOTATION AND PRELIMINARIES

### 3.1 NOTATION

For a positive integer $C$, let $[C] \triangleq \{1, \ldots, C\}$. Denote by $P[i]$ the $i$-th element of vector $P$. For two vectors $U$ and $V$, denote by $U \cdot V$ their inner product. We use $|\mathcal{C}|$ to denote the cardinality of a set $\mathcal{C}$. Also, a $C$ dimensional probability simplex is denoted by $\Delta^C$. The cross entropy of two probability distributions $P_1, P_2 \in \Delta^C$ is defined as $H(P_1, P_2) = \sum_{c=1}^{C} -P_1[c] \log P_2[c]$, and their Kullback–Leibler (KL) divergence is defined as $\mathrm{KL}(P_1||P_2) = \sum_{c=1}^{C} P_1[c] \log \frac{P_1[c]}{P_2[c]}$.

We use $\Pr(E)$ to denote the probability of event $E$. For a random variable $X$, denote by $\mathbb{P}_X$ its probability distribution, and by $\mathbb{E}_X[\cdot]$ the expected value w.r.t. $X$. For two random variables $X$ and $Y$, denote by $\mathbb{P}_{(X,Y)}$ their joint distribution. The mutual information between two random variables $X$ and $Y$ is given by $I(X, Y) = H(X) - H(X|Y)$, and the conditional mutual information of $X$ and $Y$ given a third random variable $Z$ is $I(X, Y|Z) = H(X|Z) - H(X|Y, Z)$.

### 3.2 THE NECESSITY OF ESTIMATING BCPD

In a classification task with $C$ classes, a DNN could be regarded as a mapping $f : x \rightarrow P_x$, where $x \in \mathbb{R}^d$ is an input image, and $P_x \in \Delta^C$. Denote by $y$ the correct label of $x$; then the classifier predicts $y$, as $\tilde{y} = \arg\max_{c \in [C]} P_x[c]$. As such, the error rate of $f$ is defined as $\epsilon = \Pr(\tilde{y} \neq y)$, and its accuracy is equal to $1 - \epsilon$. Assuming that $X$ and $Y$ are the random variables representing the input sample and the ground truth label, then one may learn such a classifier by minimizing the risk

$$R(f, \ell) \triangleq \mathbb{E}_{(X,Y)} \left[ \ell\left(Y, P_X\right)\right] = \mathbb{E}_X \left[ \mathbb{E}_{Y|X} \left[ \ell\left(Y, P_X\right)\right]\right] = \mathbb{E}_X \left[ (P_X^*)^T \cdot \boldsymbol{\ell}(P_X)\right], \quad (1)$$

where $\ell(\cdot)$ is the loss function and $\boldsymbol{\ell}(\cdot) \triangleq [\ell(1, \cdot), \ldots, \ell(C, \cdot)]$ is the vector of loss function, $P_X^* \triangleq [\Pr(y|X)]_{y \in [C]}$ is Bayes class probability distribution over the labels.

When using the cross entropy loss, $\ell(y, P_X) = -\mathbf{e}(y)^T \cdot \log\left(P_X\right) = -\log\left(P_X[y]\right)$, where $\mathbf{e}(y) \in \{0, 1\}^C$ denotes the *one-hot vector* of $y$. Therefore, $R(f, \ell = \mathrm{H})$ using cross entropy loss becomes

$$R(f, \mathrm{H}) = -\mathbb{E}_X \left[ (P_X^*)^T \cdot \log\left(P_X\right)\right]. \quad (2)$$

As discussed in Yang et al. (2023), the error rate of classifier $f$ is upper-bounded as

$$\epsilon \le C \times R(f, \mathbf{H}). \tag{3}$$

As such, by minimizing $R(f, \mathbf{H})$ one can decrease (increase) the classifier's error rate (accuracy). However, in a typical deep learning algorithm, both the probability density function of $x$, namely $\mathbb{P}_X$, and also $P_X^*$ are unknown. Hence, one may learn such a classifier by minimizing the *empirical risk* on a training sample $\mathcal{D} \triangleq \{(x_n, y_n)\}_{n=1}^N$ defined as:

$$R_{\text{emp}}(f, \mathbf{H}) \triangleq \frac{-1}{N} \sum_{n \in [N]} \mathbf{e}(y_n)^T \cdot \log(P_{x_n}). \tag{4}$$

By comparing eq. (2) and eq. (4), it is seen that in eq. (4): (i) $\mathbb{P}_X$ is approximated by $\frac{1}{N}$, and (ii) $P_X^*$ is approximated by $\mathbf{e}(y)$ which is an unbiased estimation of $P_X^*$. The former approximation is reasonable, however, the latter results in a significant loss in granularity. Specifically, images typically contain a wealth of information, and assigning a one-hot vector to $P_X^*$ results in a substantial loss of information. As will be discussed in the next subsection, KD alleviates this issue to some extent.

### 3.3 ESTIMATING BCPD BY THE TEACHER IN KD

In KD, the role of the teacher is to provide the student with a better estimate of $P_X^*$ compared to one-hot vectors. Denote by $P_x^t$ and $P_x^s$ the teacher's and student's outputs to sample $x$, respectively. First, the teacher is trained to minimize the empirical loss eq. (4), which is equivalent to train it to maximize the empirical LL of the ground-truth probability defined as

$$\text{LL}_{\text{emp}}(f) \triangleq \frac{1}{N} \sum_{n \in [N]} \log(P_{x_n}[y_n]). \tag{5}$$

Then, the teacher's output $P_x^t$ could be regarded as an estimate of $P_X^*$ obtained using MLL method. Afterward, the student uses the teacher's estimate of $P_X^*$, and minimizes

$$R_{\text{kd}}(f, \mathbf{H}) \triangleq \frac{-1}{N} \sum_{n \in [N]} \left(P_{x_n}^t\right)^T \cdot \log\left(P_{x_n}^s\right), \tag{6}$$

where the one-hot vector $\mathbf{e}(y_n)$ in eq. (4) is replaced by the teacher's output probability $P_{x_n}^t$.

In the next section, we aim to train the teacher such that it can provide the student with a better estimate of $P_X^*$ compared to the one that it can provide using MLL training method.

## 4 ESTIMATING BCPD VIA MCMI

In this section, we improve the teacher's estimate of the true BCPD by training it to learn more contextual information of the input images.

### 4.1 CAPTURING CONTEXTUAL INFORMATION VIA CMI

As discussed in section 1, each manifestation (a training/testing image) comprises two types of information: (i) its *prototype* (ground truth label), and (ii) some contextual information on top its *prototype*. In the context of KD, if the teacher aims to estimate the true $P_X^*$, these two types of information should be captured in its predictions. However, when a teacher is trained using MLL, it solely aims at the prototype information. Now the question is that how we can help the teacher to also learn the contextual information better.

Let $\hat{Y}$ be the label predicted by the DNN with probability $P_X[\hat{Y}]$ in response to the input $X$; meaning that for any input sample $x \in \mathbb{R}^d$ and any $\hat{y} \in [C]$, $\Pr(\hat{Y} = \hat{y} | X = x) = P_x[\hat{y}]$. The input $X$ and the probability of wrong classes have intricate non-linear relationships. For a classifier $f$, these complex relationships can be captured via $\text{CMI}(f) = I(X; \hat{Y} | Y)$. This value is the quantity of information shared between $X$ and $\hat{Y}$ when $Y$ is known.

The CMI could be also written as $I(X; \hat{Y} | Y) = H(X|Y) - H(X|\hat{Y}, Y)$, i.e., the difference between the average remaining uncertainty of $X$ when $Y$ is known and that of $X$ when both $Y$ and $\hat{Y}$ are

known. As such, maximizing $I(X; \hat{Y} \mid Y)$ is equivalent to minimizing $H(X|\hat{Y}, Y)$ (since $H(X|Y)$ is constant as it is an intrinsic characteristic of the dataset), which, in turn, forces $\hat{Y}$ to have as much information as possible about $X$, given its *prototype*.

## 4.2 CMI VALUE FOR A MODEL

As shown in Yang et al. (2023), for a classifier $f$ we have

$$I(X; \hat{Y} \mid Y = y) = \sum_{x} P_{X|Y}(x|y) \left[ \sum_{i=1}^{C} P(\hat{Y} = i|x) \times \log \frac{P(\hat{Y} = i|x)}{P_{\hat{Y}|y}(\hat{Y} = i|Y = y)} \right] \tag{7}$$

$$= \mathbb{E}_{X|Y} \left[ \left( \sum_{i=1}^{C} P_X[i] \ln \frac{P_X[i]}{P_{\hat{Y}|y}(\hat{Y} = i|Y = y)} \right) |Y = y \right] \tag{8}$$

$$= \mathbb{E}_{X|Y} \left[ \text{KL}(P_X || P_{\hat{Y}|y})|Y = y \right]. \tag{9}$$

On the other hand,

$$\text{CMI}(f) = I(X; \hat{Y} \mid Y) = \sum_{y \in [C]} P_Y(y) I(X; \hat{Y}|y). \tag{10}$$

Therefore, from eqs. (9) and (10), CMI$(f)$ could be calculated as[2]

$$\text{CMI}(f) = \mathbb{E}_{(X,Y)} \left[ \text{KL} \left( P_X || Q^Y \right) \right], \quad \text{with} \quad Q^Y \triangleq \mathbb{E}_{(X|Y)} \left[ P_X | Y \right]. \tag{11}$$

For a given dataset $\mathcal{D}$ with unknown $\mathbb{P}_{(X,Y)}$ and $\mathbb{P}_{(X|Y)}$, we can approximate CMI$(f)$ by its empirical value. To this end, first we define $\mathcal{D}_y = \{x_j \in \mathcal{D} : y_j = y\}$ the set of all training samples whose ground truth labels are $y$. Then, the empirical CMI$(f)$ could be calculated as

$$\text{CMI}_{\text{emp}}(f) = \frac{1}{N} \sum_{y \in [C]} \sum_{x_j \in \mathcal{D}_y} \text{KL}(P_{x_j} || Q^y_{\text{emp}}), \tag{12}$$

$$\text{where} \quad Q^y_{\text{emp}} = \frac{1}{|\mathcal{D}_y|} \sum_{x_j \in \mathcal{D}_y} P_{x_j}, \quad \text{for } y \in [C]. \tag{13}$$

Deploying eq. (12), in the next subsection we explain that the role of temperature $T$ in KD is to naively increase the teacher's CMI value.

## 4.3 THE ROLE OF TEMPERATURE IN KD

In KD, using a higher temperature $T$ encourages the student to focus also on the wrong labels' logits, inside which the dark knowledge of the teacher resides (Hinton et al., 2015). Although some prior works tried to optimize $T$ as a hyper-parameter of the training algorithm (Li et al., 2023; Liu et al., 2022; Jafari et al., 2021; Stanton et al., 2021), the real physical meaning of $T$ has yet to be elucidated.

We argue that $T$ is a simple means to gauge the teacher's CMI value to some extent. To demonstrate this, fig. 1 depicts the teacher's CMI value (right vertical axis) along with the student's classification accuracy (left vertical axis) Vs. the teacher's temperature (horizontal axis) for three different teacher-student pairs trained on CIFAR-100 dataset, where we use $T = \{1, 2, \ldots, 8\}$ (here, following Hinton et al. (2015), both teacher and student models use the same T value). Let us, for instance, consider fig. 1a. As seen, by increasing the $T$ value, the teacher's CMI initially increases reaching its maximum value at $T^* = 2$, and then it continually drops. Interestingly, for $T^* = 2$, the student's accuracy reaches its highest value. The similar conclusions can be made from figs. 1b and 1c suggesting that $T$ is used in KD as a means of increasing the teacher's CMI value.

## 5 METHODOLOGY

Based on the discussions provided thus far, in order for the teacher to predict the true BCPD well, it shall maximize the (i) log-likelihood of the ground truth label, and (ii) its CMI at the same time.

---

[2]To understand how CMI(f) affects the output probability space of $f$, please refer to appendix A.4.

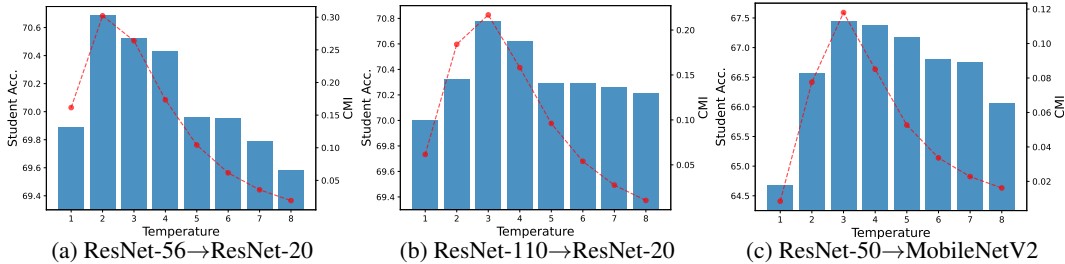

(a) ResNet-56→ResNet-20    (b) ResNet-110→ResNet-20    (c) ResNet-50→MobileNetV2

Figure 1: The teacher's CMI value (red curve, right axis) along with the student's accuracy (blue bars, left axis) Vs. the teacher's temperature in conventional KD for three different teacher-student pairs.

Hence, we train the teacher to **maximize** the following objective function

$$\ell_{\text{MCMI}}(f) = \text{LL}_{\text{emp}}(f) + \lambda\,\text{CMI}_{\text{emp}}(f), \tag{14}$$

where $\lambda > 0$ is a hyper-parameter. We refer to such a training framework as MCMI estimation. Yet, maximizing $\text{CMI}_{\text{emp}}(f)$ faces some challenges as the term $Q^y_{\text{emp}}$ in $\text{CMI}_{\text{emp}}(f)$ is a function of $P_{x_j}$ (see eq. (12)). Hence, the optimization problem in eq. (14) is not amenable to numerical solutions. To tackle this issue, instead of training the teacher from scratch, we fine-tune a pre-trained teacher in the manner explained in the following.

First we pick a pre-trained teacher model that is trained via cross entropy loss. Then, we find $\{Q^y_{\text{emp}}\}_{y\in[C]}$ for this model using eq. (13). Then, we fine-tune the pre-trained model by maximizing the objective function in eq. (14) while keeping the $\{Q^y_{\text{emp}}\}_{y\in[C]}$ values constant over the course of fine-tuning.

**Remark.** *One may wonder why training a teacher using traditional MLL is not suitable for KD, and why teachers trained via early stopping are better teachers (Cho & Hariharan, 2019). To answer to these questions, we plot the curves for CMI and LL values for two models trained on CIFAR-100 using MLL loss in fig. 2. As seen, the teacher's CMI starts to sharply decrease after epoch ∼150. Therefore, using an early-stopped teacher is beneficial since it has a higher CMI value compared to the fully converged one. Yet, the problem with an early-stopped teacher is that it has a low LL value. This issue does not exist for a model trained by MCMI as both CMI and LL are maximized[3].*

**Proposition 1.** *If $\boldsymbol{f}_x$ is an intermediate feature map of a DNN corresponding to the input $x$, then, $I(X;\hat{Y}\mid Y) \le I(X;\boldsymbol{f}_X\mid Y)$*[4].

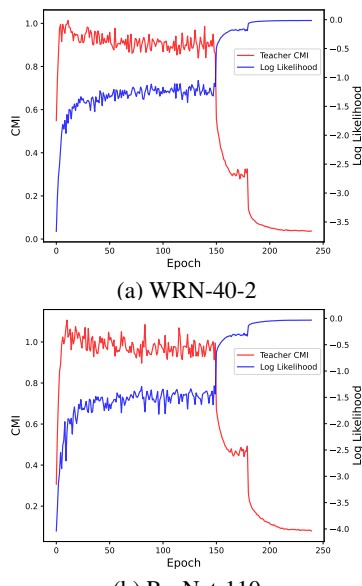

(a) WRN-40-2

(b) ResNet-110

Figure 2: The evolution of CMI and LL values for a teacher trained by MLL during the training.

Based on proposition 1, maximizing $I(X;\hat{Y}\mid Y)$, would also have the effect of increasing $I(X;\boldsymbol{f}_X\mid Y)$. Therefore, no matter what kind of KD method is used, logit-based or representation-based, maximizing the loss $\ell_{\text{MCMI}}(f)$ would enable the teacher to provide more contextual information to the student.

## 6 EXPERIMENTS

• **Terminologies**: For the sake of brevity, hereafter, we refer to the teachers trained by MLL and MCMI methods as the "MLL teacher" and "MCMI teacher", respectively.

In this section, we conclude the paper with several experiments to demonstrate the superior effectiveness of MCMI estimator compared to its MLL counterpart. Specifically, in section 6.1, we show that replacing the MLL teacher with the MCMI one in the state-of-the-art KD methods leads to a

---

[3]For a comparison between the MCMI and early-stopped teacher, please refer to appendix A.3.

[4]See appendix A.5 for the proof.

consistent increase in the student's accuracy. In sections 6.2 and 6.3, we demonstrate that the MCMI teacher can drastically increase the student's accuracy in the zero-shot and few-shot settings. In addition, in appendix A.2, we justify why larger teachers often harm the student's accuracy, and show that this problem is resolved by using MCMI teachers.

• **Plug-and-play nature of MCMI teacher**: In all the experiments conducted in this section, when testing the performance of the MCMI teacher, we do not tune any hyper-parameters in the underlying knowledge transfer methods, all of which are the same as in the corresponding benchmark methods.

## 6.1 MCMI TEACHER IN KD METHODS

In order to demonstrate that $P^t_{x,\text{MCMI}}$ serves as a better estimate of $P^*_X$ compared to $P^t_{x,\text{MLL}}$, in appendix A.6, we perform experiments on a synthetic dataset with a known $P^*_X$. Additionally, a litmus test to see whether $P^t_{x,\text{MCMI}}$ outperforms $P^t_{x,\text{MLL}}$ as an estimate of $P^*_X$ is the student's accuracy; where a higher student's accuracy indicates a better estimate provided by the teacher. In the following, we conduct extensive experiments over CIFAR-100 (Krizhevsky et al., 2009) and ImageNet (Russakovsky et al., 2015) datasets, showing the MCMI teacher's effectiveness.

• **CIFAR-100.** This dataset contains 50K training and 10K test colour images of size $32 \times 32$, which are labeled for 100 classes. Following the settings of CRD (Tian et al., 2019), we employ 6 teacher-student pairs with identical network architectures and another 6 pairs with differing network architectures for our experiments (see table 1). We perform each experiment across 5 independent runs, and report the average accuracy (for the accuracy variances, refer to appendix A.12).

For comprehensive comparisons, we compare using a MCMI teacher Vs. MLL teacher in the existing state-of-the-art distillation methods, including KD (Hinton et al., 2015), AT (Zagoruyko & Komodakis, 2016), PKT (Passalis & Tefas, 2018), SP (Tung & Mori, 2019), CC (Peng et al., 2019), RKD (Park et al., 2019a), VID (Ahn et al., 2019), CRD (Tian et al., 2019), DKD (Zhao et al., 2022), REVIEWKD Chen et al. (2021a), and HSAKD (Yang et al., 2021). All the training setups, including fine-tuning MCMI teachers, training students, and $\lambda$ values, along with a study on the effect of $\lambda$ on MCMI teachers are provided in appendix A.8.

As observed in table 1, whether the teacher-student architectures are the same or different, the student's accuracy is improved across the board by replacing the MLL teacher with the MCMI one. In addition, such improvements are consistent in all the tested KD methods, and could be up to $3.32\%$; nevertheless, the gain is more notable when the teacher and student have a different architecture. For both MLL and MCMI teachers, we reported their accuracies over the testing set, and both LL and CMI values over the training set in table 1. As seen, the test accuracy of the MCMI teacher is always lower than that of the MLL teacher, confirming the fact that training the teacher in KD for the benefit of the student is a different task from training the teacher for its own prediction accuracy.

• **ImageNet.** ImageNet is a large-scale dataset used in visual recognition tasks, containing around 1.2 million training and 50K validation images. Following the settings of (Tian et al., 2019; Yang et al., 2020), we use 2 popular teacher-student pairs for our experiments (see table 2). We note that across all the knowledge transfer methods reported in table 2, replacing the MLL teacher by MCMI teacher consistently leads to an increase in the student accuracy. For example, when considering teacher-student pairs ResNet34-ResNet18 and ResNet50-MobileNetV2, the increase in the student's Top-1 accuracy in KD is 0.51% and 0.60%, respectively.

• **Visualization of contextual information extracted by MCMI teachers.** In this subsection, we use Eigen-CAM (Bany Muhammad & Yeasin, 2021) to visualize and compare the feature maps extracted by MCMI and MLL teachers. To this end, we randomly pick four images from the same class in ImageNet dataset, namely "Toy Terrier", and depict the respective MCMI and MLL teacher's Eigen-CAM for them (see fig. 3). As seen, the activation maps provided by the MLL teacher always highlight the dog's head. However, for the MCMI teacher, the activation maps

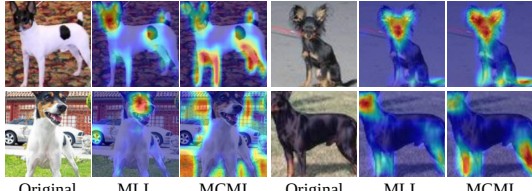

Original   MLL   MCMI   Original   MLL   MCMI

Figure 3: Eigen-CAM for MLL and MCMI teachers for 4 samples from class "Toy Terrier".

Table 1: The test accuracy (%) of student networks on CIFAR-100 (averaged over 5 runs), with teacher-student pairs of the same/different architectures. The subscript denotes the improvement achieved by replacing MLL teacher with MCMI teacher. We use **bold** numbers and asterisk ($^*$) to denote the best results and to identify the results reproduced on our local machines, respectively.

**Teachers and students with the same architectures.**

| Teacher | ResNet-56 MLL | ResNet-56 MCMI | ResNet-110 MLL | ResNet-110 MCMI | ResNet-110 MLL | ResNet-110 MCMI | WRN-40-2 MLL | WRN-40-2 MCMI | WRN-40-2 MLL | WRN-40-2 MCMI | VGG-13 MLL | VGG-13 MCMI |
|---|---|---|---|---|---|---|---|---|---|---|---|---|
| Accuracy | 72.34 | 72.09 | 74.31 | 73.60 | 74.31 | 73.60 | 75.61 | 75.21 | 75.61 | 75.21 | 74.64 | 73.96 |
| LL | -0.101 | -0.344 | -0.033 | -0.246 | -0.033 | -0.246 | -0.052 | -0.132 | -0.052 | -0.132 | -0.007 | -0.076 |
| CMI | 0.1583 | 0.4428 | 0.0605 | 0.3474 | 0.0605 | 0.3474 | 0.0255 | 0.1951 | 0.0255 | 0.1951 | 0.0152 | 0.1298 |
| Student | ResNet-20 | | ResNet-20 | | ResNet-32 | | WRN-16-2 | | WRN-40-1 | | VGG-8 | |
| Accuracy | 69.06 | | 69.06 | | 71.14 | | 73.26 | | 71.98 | | 70.36 | |
| KD | 70.66 | 70.84 $_{+0.18}$ | 70.67 | 70.85 $_{+0.18}$ | 73.08 | 73.48 $_{+0.40}$ | 74.92 | 75.42 $_{+0.50}$ | 73.54 | 74.53 $_{+0.99}$ | 72.98 | 73.83 $_{+0.85}$ |
| AT | 70.55 | 70.89 $_{+0.34}$ | 70.22 | 70.68 $_{+0.46}$ | 72.31 | 73.96 $_{+1.65}$ | 74.08 | 74.49 $_{+0.41}$ | 72.77 | 73.25 $_{+0.48}$ | 71.43 | 71.76 $_{+0.33}$ |
| PKT | 70.34 | 70.96 $_{+0.62}$ | 70.25 | 71.03 $_{+0.78}$ | 72.61 | 72.92 $_{+0.31}$ | 74.54 | 75.01 $_{+0.47}$ | 73.45 | 74.15 $_{+0.70}$ | 72.88 | 73.35 $_{+0.47}$ |
| SP | 69.67 | 70.98 $_{+1.31}$ | 70.04 | 70.83 $_{+0.79}$ | 72.69 | 73.34 $_{+0.65}$ | 73.83 | 74.60 $_{+0.77}$ | 72.43 | 73.60 $_{+1.17}$ | 72.68 | 73.29 $_{+0.61}$ |
| CC | 69.63 | 69.98 $_{+0.35}$ | 69.48 | 70.02 $_{+0.54}$ | 71.48 | 71.71 $_{+0.23}$ | 73.56 | 74.00 $_{+0.44}$ | 72.21 | 72.50 $_{+0.29}$ | 70.71 | 71.02 $_{+0.31}$ |
| RKD | 69.61 | 70.68 $_{+1.07}$ | 69.25 | 70.24 $_{+0.99}$ | 71.82 | 72.65 $_{+0.83}$ | 73.35 | 73.97 $_{+0.62}$ | 72.22 | 72.66 $_{+0.44}$ | 71.48 | 72.03 $_{+0.55}$ |
| VID | 70.38 | 70.64 $_{+0.26}$ | 70.16 | 70.69 $_{+0.53}$ | 72.61 | 73.10 $_{+0.49}$ | 74.11 | 74.44 $_{+0.33}$ | 73.30 | 73.58 $_{+0.28}$ | 71.23 | 71.93 $_{+0.70}$ |
| CRD | 71.16 | 71.40 $_{+0.24}$ | 71.46 | 71.93 $_{+0.47}$ | 73.48 | 74.03 $_{+0.55}$ | 75.48 | 75.82 $_{+0.34}$ | 74.14 | 74.86 $_{+0.72}$ | 73.94 | 74.23 $_{+0.29}$ |
| DKD | 71.97 | 72.31 $_{+0.34}$ | 71.51* | 71.83 $_{+0.32}$ | 74.11 | 74.36 $_{+0.25}$ | 76.24 | 76.66 $_{+0.42}$ | 74.81 | 75.63 $_{+0.82}$ | 74.68 | 74.87 $_{+0.19}$ |
| REVIEW | 71.89 | 72.31 $_{+0.42}$ | 71.65* | 72.11 $_{+0.46}$ | 73.89 | 74.01 $_{+0.12}$ | 76.12 | 76.29 $_{+0.17}$ | 75.09 | 75.47 $_{+0.38}$ | 74.84 | 74.96 $_{+0.12}$ |
| HSAKD | 72.58 | **72.70** $_{+0.12}$ | 72.64* | **73.15** $_{+0.51}$ | 74.97* | **75.71** $_{+0.74}$ | 77.20 | **77.36** $_{+0.16}$ | 77.00 | **77.55** $_{+0.55}$ | 75.42* | **75.86** $_{+0.44}$ |

**Teachers and students with different architectures.**

| Teacher | ResNet-50 MLL | ResNet-50 MCMI | ResNet-50 MLL | ResNet-50 MCMI | ResNet-32×4 MLL | ResNet-32×4 MCMI | ResNet-32×4 MLL | ResNet-32×4 MCMI | WRN-40-2 MLL | WRN-40-2 MCMI | VGG-13 MLL | VGG-13 MCMI |
|---|---|---|---|---|---|---|---|---|---|---|---|---|
| Accuracy | 79.34 | 78.45 | 79.34 | 78.45 | 79.41 | 78.70 | 79.41 | 78.70 | 75.61 | 75.21 | 74.64 | 73.96 |
| LL | -0.004 | -0.083 | -0.004 | -0.083 | -0.004 | -0.035 | -0.004 | -0.035 | -0.052 | -0.132 | -0.007 | -0.076 |
| CMI | 0.0085 | 0.1065 | 0.0085 | 0.1065 | 0.0059 | 0.0586 | 0.0059 | 0.0586 | 0.0255 | 0.1951 | 0.0152 | 0.1298 |
| Student | MobileNetV2 | | VGG-8 | | ShuffleNetV1 | | ShuffleNetV2 | | ShuffleNetV1 | | MobileNetV2 | |
| Accuracy | 64.60 | | 70.36 | | 70.50 | | 71.82 | | 70.50 | | 64.60 | |
| KD | 67.35 | 70.23 $_{+2.88}$ | 73.81 | 74.59 $_{+0.78}$ | 74.07 | 75.90 $_{+1.83}$ | 74.45 | 76.32 $_{+1.87}$ | 74.83 | 76.45 $_{+1.62}$ | 67.37 | 69.14 $_{+1.77}$ |
| AT | 58.58 | 60.03 $_{+1.45}$ | 71.84 | 72.19 $_{+0.35}$ | 71.73 | 75.05 $_{+3.32}$ | 72.73 | 75.21 $_{+2.48}$ | 73.32 | 75.61 $_{+2.29}$ | 59.40 | 62.07 $_{+2.67}$ |
| PKT | 66.52 | 67.42 $_{+0.90}$ | 73.10 | 73.43 $_{+0.33}$ | 74.10 | 75.21 $_{+1.11}$ | 74.69 | 76.34 $_{+1.65}$ | 73.89 | 75.39 $_{+1.50}$ | 67.13 | 68.37 $_{+1.24}$ |
| SP | 68.08 | 69.07 $_{+0.99}$ | 73.34 | 74.14 $_{+0.80}$ | 73.48 | 76.56 $_{+3.08}$ | 74.56 | 76.70 $_{+2.14}$ | 74.52 | 76.82 $_{+2.30}$ | 66.30 | 67.83 $_{+1.53}$ |
| CC | 65.43 | 66.76 $_{+1.33}$ | 70.25 | 70.90 $_{+0.65}$ | 71.14 | 71.77 $_{+0.63}$ | 71.29 | 73.02 $_{+1.73}$ | 71.38 | 71.80 $_{+0.42}$ | 64.86 | 65.45 $_{+0.59}$ |
| RKD | 64.43 | 65.11 $_{+0.68}$ | 71.50 | 72.10 $_{+0.60}$ | 72.28 | 73.59 $_{+1.31}$ | 73.21 | 74.67 $_{+1.46}$ | 72.21 | 74.26 $_{+2.05}$ | 64.52 | 65.37 $_{+0.85}$ |
| VID | 67.57 | 67.61 $_{+0.04}$ | 70.30 | 70.69 $_{+0.39}$ | 73.38 | 74.58 $_{+1.20}$ | 73.40 | 74.67 $_{+1.27}$ | 73.61 | 75.03 $_{+1.42}$ | 65.56 | 65.77 $_{+0.21}$ |
| CRD | 69.11 | 69.70 $_{+0.59}$ | 74.30 | 74.86 $_{+0.56}$ | 75.11 | 76.82 $_{+1.71}$ | 75.65 | 77.54 $_{+1.89}$ | 76.05 | 76.62 $_{+0.57}$ | 69.70 | 69.98 $_{+0.28}$ |
| DKD | 70.35 | 71.70 $_{+1.35}$ | 73.94* | 75.35 $_{+1.41}$ | 76.45 | 77.21 $_{+0.76}$ | 77.07 | 77.66 $_{+0.59}$ | 76.70 | 77.42 $_{+0.72}$ | 69.71 | 70.35 $_{+0.64}$ |
| REVIEW | 69.89 | 70.63 $_{+0.74}$ | 73.43* | 74.20 $_{+0.77}$ | 77.45 | 77.78 $_{+0.33}$ | 77.78 | 78.23 $_{+0.45}$ | 77.14 | 77.56 $_{+0.42}$ | 70.37 | 71.70 $_{+1.33}$ |
| HSAKD | 71.83* | **72.98** $_{+1.15}$ | 75.87* | **76.45** $_{+0.58}$ | 79.51* | **79.77** $_{+0.26}$ | 79.93 | **80.01** $_{+0.08}$ | 78.51 | **78.87** $_{+0.36}$ | 71.09* | **72.80** $_{+1.71}$ |

Table 2: Top-1 and Top-5 student's test accuracy (%) on ImageNet validation set for 4 different KD methods using MLL and MCMI teachers (RN and MN stand for ResNet and MobileNet, respectively).

| Teacher-Student | | Teacher Performance Top1 | Top5 | CMI | LL | KD Top1 | Top5 | DKD Top1 | Top5 | ReviewKD Top1 | Top5 | CRD Top1 | Top5 |
|---|---|---|---|---|---|---|---|---|---|---|---|---|---|
| RN34-R18 | MLL | 73.31 | 91.42 | 0.7203 | -0.5600 | 71.03 | 90.05 | 71.70 | 90.41 | 71.61 | 90.51 | 71.17 | 90.13 |
| | MCMI | 71.62 | 90.67 | 0.8650 | -0.6262 | 71.54 | 90.63 | **72.06** | 91.12 | 71.98 | 90.86 | 71.50 | 90.20 |
| RN50-MNV2 | MLL | 76.13 | 92.86 | 0.6002 | -0.4492 | 70.50 | 89.80 | 72.05 | 91.05 | 72.56 | 91.00 | 71.37 | 90.41 |
| | MCMI | 74.94 | 91.03 | 0.7150 | -0.4943 | 71.10 | 90.16 | 72.61 | 91.26 | **73.00** | 92.19 | 71.63 | 90.53 |

as a whole capture more contextual information, including the dog's legs and some surrounding information (for a comprehensive illustration of the Eigen-CAM figures, refer to appendix A.10).

## 6.2 ZERO-SHOT CLASSIFICATION IN KD

In the concept of zero-shot classification in KD, the teacher is trained on the entire dataset, but the samples for some of the classes are completely omitted for the student during the distillation. Then, the student is tested against the samples for the whole dataset (both seen and unseen classes). Some real-world scenarios of this setup is elaborated in appendix A.11.1. To show the effectiveness of using MCMI teacher in KD for zero-shot classification, here we report an example of such scenario on CIFAR-10 dataset, and further results including those for CIFAR-100 are reported in appendix A.11.4. We use ResNet56-ResNet20 as the teacher-student pair, and train the teacher via MLL and MCMI methods on the whole dataset. Then, we entirely omit five classes from the student's training set, and train it once using MLL teacher and once using MCMI teacher. The student's confusion matrices[5] for both scenarios are depicted in fig. 4 where the dropped classes are highlighted in red. As seen, the student's accuracies for the dropped classes are always zero when using the MLL teacher. On the other hand, these accuracies are substantially increased, by as much as 84%, when using the MCMI teacher. We conducted additional tests in which we dropped varying numbers of classes for the student, as elaborated in A.11.4.

## 6.3 FEW-SHOT CLASSIFICATION

In few-shot classification, the models are provided with an $\alpha$ percent of the instances in each class Chen et al. (2018). Translating this concept into the context of KD, the teacher is trained on the full dataset, yet an $\alpha$ percent of the samples in each class is used to train the student during the distillation process. Here, we show the effectiveness of MCMI teacher over MLL teacher using the following experiments on

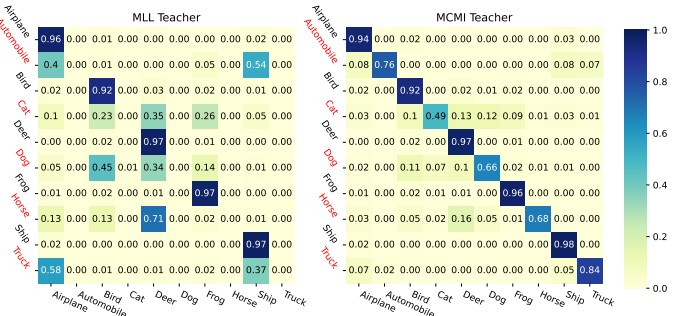

Figure 4: The student's confusion matrices when it is trained through the MLL teacher (left), and by MCMI teacher (right).

CIFAR-100. We use KD and CRD with ResNet-56 and ResNet-20 as the teacher and student models. We conducted experiments for different values of $\alpha$, namely $\{5, 10, 15, 25, 35, 50, 75\}$. The results are summarized in table 3, where for a fair comparison, we employ an identical partition of the training set for each few-shot setting. As seen, the improvement in the student's accuracy is notable by using an MCMI teacher, which is particularly more pronounced for smaller $\alpha$ values (for instance, when $\alpha = 5$, the improvement in the student's accuracy is $5.72\%$ and $3.8\%$ when KD and CRD are used, respectively).

Table 3: Top-1 accuracy (%) of the student under few-shot setting (averaged over 5 runs).

| $\alpha$ | 5 | | 10 | | 15 | | 25 | | 35 | | 50 | | 75 | |
|---|---|---|---|---|---|---|---|---|---|---|---|---|---|---|
| Teacher | MLL | MCMI | MLL | MCMI | MLL | MCMI | MLL | MCMI | MLL | MCMI | MLL | MCMI | MLL | MCMI |
| KD | 52.30 | 58.02 +5.72 | 60.13 | 63.75 +3.62 | 63.52 | 66.20 +2.68 | 66.78 | 68.10 +1.32 | 68.28 | 69.34 +1.06 | 69.52 | 70.28 +0.76 | 70.44 | 70.59 +0.15 |
| CRD | 47.60 | 51.40 +3.80 | 54.60 | 56.80 +2.20 | 58.90 | 60.02 +1.12 | 63.82 | 64.7 +0.88 | 66.70 | 67.22 +0.52 | 68.84 | 69.15 +0.31 | 70.35 | 70.40 +0.05 |

## 7 CONCLUSION

In conclusion, this paper has introduced a novel approach, the maximum conditional mutual information (MCMI) method, to enhance knowledge distillation (KD) by improving the estimation of the Bayes conditional probability distribution (BCPD) used in student training. Unlike conventional methods that rely on maximum log-likelihood (MLL) estimation, MCMI simultaneously maximizes both log-likelihood and conditional mutual information (CMI) during teacher training. This approach effectively captures contextual image information, as visualized through Eigen-CAM. Extensive experiments across various state-of-the-art KD frameworks have demonstrated that utilizing a teacher trained with MCMI leads to a consistent increase in student's accuracy.

---

[5]Please refer to appendix A.11.4 for a detailed explanation of how to interpret the confusion matrix.

## 8 ACKNOWLEDGMENTS

We would like to thank the three anonymous reviewers for spending time and efforts and bringing in constructive questions and suggestions, which help us greatly to improve the quality of the paper. We would like to also thank the Program Chairs and Area Chairs for handling this paper and providing the valuable and comprehensive comments.

This work was supported in part by the Natural Sciences and Engineering Research Council of Canada under Grant RGPIN203035-22, and in part by the Canada Research Chairs Program.

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

# A APPENDIX

## A.1 RELATED WORKS ON MUTUAL INFORMATION BASED KD

Viewed through the lens of information theory, knowledge distillation can be regarded as the process of maximizing the mutual information between teacher and student models. Considering that the true distribution between the teacher and the student representations are unknown, Ahn et al. (2019) maximizes a variational lower-bound for the mutual information between the teacher and the student representations by approximating an intractable conditional distribution using a pre-defined variational distribution. To better retain sufficient and task-relevant knowledge, Tian et al. (2021) utilized the information bottleneck (Tishby et al., 2000) to produce highly-represented encodings. In order to encompass structural knowledge, specifically, the high-order relationships among sample representations, Zhu et al. (2021); Tian et al. (2019) have introduced an approach that involves maximizing the lower bound of mutual information between the anchor-teacher relation and the anchor-student relation. This approach offers a practical solution within the realm of contrastive learning. Shrivastava et al. (2023) employs a contrastive objective, allowing to simultaneously estimate and maximize a lower bound on the mutual information pertaining to local and global feature representations shared between a teacher and a student network. Furthermore, Passalis et al. (2020); Passalis & Tefas (2018) employed the Quadratic Mutual Information (Torkkola, 2003) (that replaces the KL divergence with a quadratic divergence) to measure the information contained within teacher and student networks.

We note that our approach differs from these previous works in that we employ information-theoretic principles primarily to optimize teacher training, as opposed to utilizing them solely for the distillation process. Indeed, our approach aids the teacher in achieving a more accurate estimation of the true BCPD.

## A.2 WHY LARGER TEACHERS HARM KD

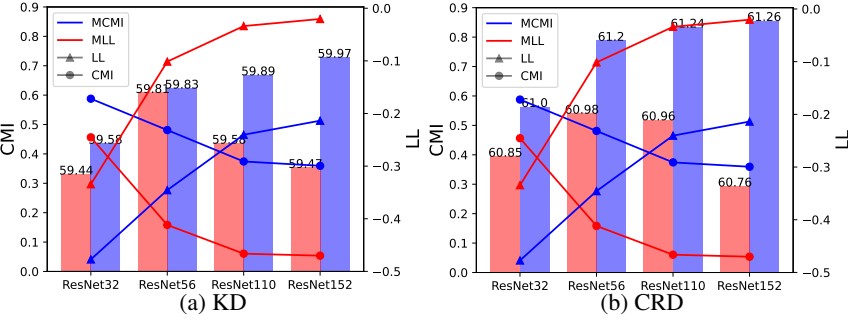

Figure 5: The effect of teacher's size on the (i) student's accuracy, (ii) teacher's CMI, and (iii) teacher's LL for both MLL and MCI teachers. The student model is ResNet8, and the dataset is CIFAR-100.

It is known that using a large model as the teacher would considerably degrade the student performance (Liu et al., 2021). Many papers attribute the cause of this issue to the *capacity gap* between the teachers and the students, and to that the small-scale model struggles to grasp the intricate, high-level semantics extracted by the larger model (Cho & Hariharan, 2019; Mirzadeh et al., 2020). Nevertheless, these explanations lack persuasiveness. In fact, we can answer to this phenomenon by observing the LL and CMI values for teachers of different sizes.

In particular, consider fig. 5a, where KD method is used to train ResNet8 using MLL and MCMI teachers of different sizes. As observed, for both MLL and MCMI teachers, as the model size increases, on the one hand, the teacher's LL value increases (beneficial), and on the other hand, the teacher's CMI decreases (detrimental). However, for MLL teachers with larger size, the CMI values are rather small, although their LL values are quite big. This is the reason that why larger MLL teachers are not often hurt the student's accuracy (as seen the student's accuracy starts decreasing as the MLL teacher model size goes beyond ResNet56).

However, this issue is mitigated when using MCMI teachers, in that the student's accuracy always rises as the teacher's size increases. This is due to the fact that compared to the MLL teacher, the CMI value for the MCMI teacher does not drop significantly as the model size increases. Therefore, an MCMI teacher with a large size can establish a better trade-off between its LL and CMI values yielding a better BCPD estimate.

### A.3 COMPARISON BETWEEN MCMI AND EARLY-STOPPED TEACHER IN KD

In this subsection, we compare the effectiveness of using MCMI teacher Vs. early-stopped (ES) one in KD. In particular, we conduct experiments on ImageNet dataset, where the teacher-student pairs are ResNet34-ResNet18, and ResNet50-MobileNetV2. The setup for training MCMI teacher is the same as that used for generating the results in table 2. For the ES teacher, following Cho & Hariharan (2019), we train ResNet34 for 50 epochs, and ResNet50 for 35 epochs. The results are summarized in table 4.

As seen, the student's accuracy is higher when it is trained by MCMI teacher compared to when it is trained by the other counterpart teachers. In fact, the problem with ES and MLL teachers are that (i) the ES teacher has a low LL value and a relatively high CMI value (to see why an ES teacher has a higher CMI value compared to MLL teacher, please refer to Yang et al. (2023)), and (ii) the MLL teacher has a high LL value, yet a low CMI value. Therefore, none-of these methods can establish a fair trade-off between MLL and LL, and consequently they cannot properly estimate the true $P_X^*$. This issue is alleviated for the MCMI teachers.

Table 4: Student's accuracy (%) on ImageNet when it is trained by MLL, ES, and CMI teachers.

| Teacher-Student | R34-R18 | | | | R50-MNV2 | | | |
|---|---|---|---|---|---|---|---|---|
| | Teacher's Acc. | CMI | LL | Student's Acc. | Teacher's Acc. | CMI | LL | Student's Acc. |
| MLL | 73.31 | 0.7203 | -0.5600 | 71.03 | 76.13 | 0.6002 | -0.4492 | 70.50 |
| ES | 68.68 | 0.9958 | -0.9217 | 71.25 | 70.14 | 0.9598 | -0.8840 | 70.76 |
| MCMI | 71.62 | 0.8650 | -0.6262 | 71.54 | 74.94 | 0.7150 | -0.4943 | 71.10 |

### A.4 HOW CMI AFFECTS THE OUTPUT SPACE OF A DNN

#### A.4.1 VISUALIZE THE OUTPUT PROBABILITY SPACE OF DNNS WITH DIFFERENT CMI VALUE

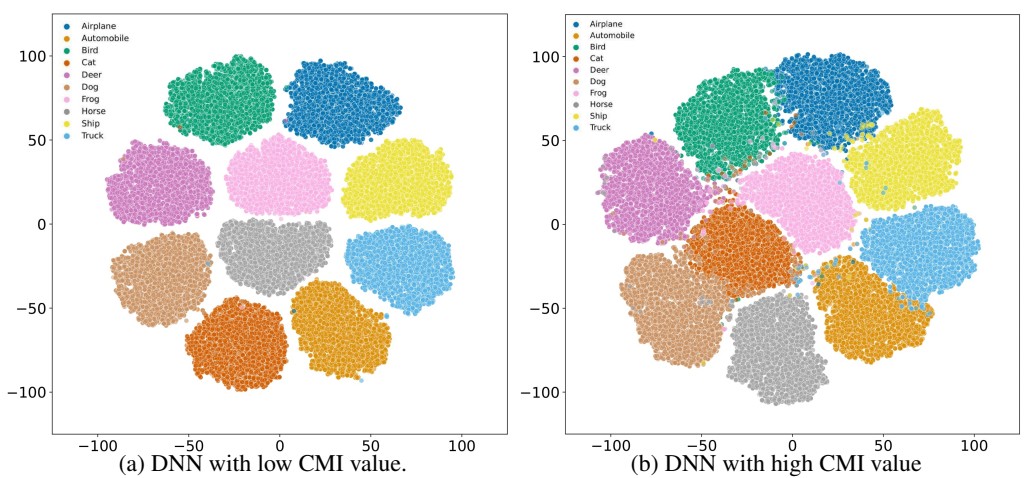

(a) DNN with low CMI value.  (b) DNN with high CMI value

Figure 6: Visualization of the output probability space of the DNN with (a) low CMI value, and (b) high CMI value.

The DNN's outputs to the instances from a specific class $Y = y$, for $y \in [C]$, form a cluster in the DNN's output space. The centroid of this cluster is the average of $P_X$ w.r.t. the conditional distribution $\mathbb{P}_{(X|Y)}(\cdot|Y = y)$, which is indeed the definition of $Q^c$ in eq. (11).

Referring to eq. (11), fixing the label $Y = y$, the $\mathrm{CMI}(f, Y = y)$ measures the average distance between the centroid $Q^y$ and the output probability vectors $P_X$ for class $y$. Therefore, $\mathrm{CMI}(f, Y = y)$ tells how all output probability distributions $P_X$ given $y$ are concentrated around its centroid $Q^y$. As such, the smaller $\mathrm{CMI}(f, Y = y)$ is, the more concentrated all output probability distributions $P_X$ given $y$ are around its centroid.

To visualize this, we train two DNNs on CIFAR-10 dataset, one with high and one with low CMI values. Then, we use t-SNE (Van der Maaten & Hinton, 2008) to map the 10-dimensional output probability space of the DNN into 2-dimensional space. The result is depicted in fig. 6. As seen, the clusters for the DNN with lower CMI value are more concentrated (less dispersed).

### A.4.2 COMPARING TEACHERS TRAINED BY THREE REGULARIZATION TERMS: CMI, ENTROPY, LABEL SMOOTHING

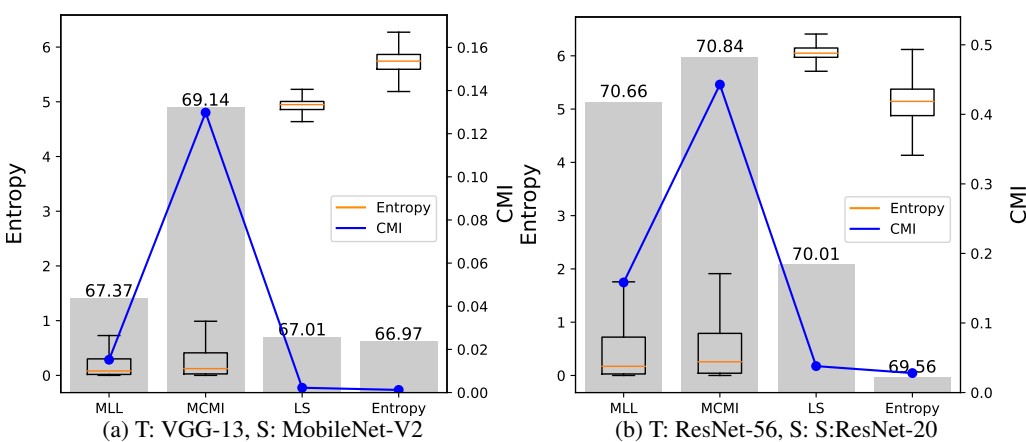

(a) T: VGG-13, S: MobileNet-V2          (b) T: ResNet-56, S: S:ResNet-20

Figure 7: The effect of various regularization methods and MCMI method on (i) student's accuracy, (ii) teacher's entropy, (iii) teacher's CMI.

In this subsection, in order to unveil the reason why label smoothing hurts student's accuracy in knowledge distillation (Tang et al., 2020), we conduct a comparative study to see how training teachers with different regularization terms, namely {CMI, entropy, label smoothing}, affect KD.

To this end, we train teachers using these regularization methods, and report the teacher's CMI value and the average entropy of its output probability vectors. Then, we use these three teachers to train a student via KD framework, and also report the accuracy of the resulting students. The results for two teacher-student pairs are depicted in fig. 7, where we also depicted the results for an MLL teacher (the teacher with no regularization term).

As observed, the CMI regularization does not increase the entropy of the teacher's output predictions. On the other hand, the teachers trained by entropy and LS regularization have low CMI value, and high output's entropy. This shows that the CMI value is not related to the entropy of the output probability vectors. In addition, we see that to train students with high accuracy, the teacher's CMI value should be high, and the entropy of its output is not important.

### A.5 PROOF OF PROPOSITION 1

*Proof.* For a given DNN, we use $\boldsymbol{f}_X$ to represent the intermediate feature to sample $X$. As discussed in Tishby & Zaslavsky (2015); Ye et al. (2022), the layered structure of a DNN forms a Markov chain as:

$$X \to \boldsymbol{f}_X \to \hat{Y}, \tag{15}$$

where $\hat{Y}$ is the output of DNN to the input $X$. By writing the CMI in terms of entropy we obtain

$$I(X; \hat{Y} \,|\, Y) = H(X|Y) - H(X|\hat{Y}, Y) \tag{16}$$

$$I(X; \boldsymbol{f}_X \,|\, Y) = H(X|Y) - H(X|\boldsymbol{f}_X, Y) \tag{17}$$

On the other hand,

$$H(X|\boldsymbol{f}_X, Y) = H(X|\boldsymbol{f}_X, \hat{Y}, Y) \tag{18}$$
$$\leq H(X|\hat{Y}, Y), \tag{19}$$

where eq. (18) is due to the properties of Markov chain, and eq. (19) holds since removing a random variable from condition increases the entropy. This together with eqs. (16) and (17) complete the proof of the proposition. □

### A.6 SYNTHETIC GAUSSIAN DATASET

First, let us denote the predictions of the MLL and MCMI teachers by $P^t_{x,\text{MLL}}$ and $P^t_{x,\text{MCMI}}$, respectively. In this subsection, we aim to examine whether $P^t_{x,\text{MCMI}}$ is a better estimate of $P^*_X$ than $P^t_{x,\text{MLL}}$.

Since the true BCPD is unknown for the popular datasets, we generate a synthetic dataset whose BCPD is known in the manner explained in the sequel (following Ren et al. (2021)).

We generate a 10-class toy Gaussian dataset with $10^5$ data points. The dataset is divided into training, validation, and test sets with a split ratio $[0.9, 0.05, 0.05]$. The underlying model (for both teacher and student) in this set of experiments is a 3-layer MLP with ReLU activation, and the hidden size is 128 for each layer. We set the learning rate as $5 \times 10^{-4}$, the batch size as 32, and the number of training epochs is 100.

The sampling process is implemented as follows: we first choose the label $y$ using a uniform distribution across all the 10 classes. Next, we sample $x|_{y=k} \sim \mathcal{N}\left(\mu_k, \sigma^2 I\right)$ as the input signal. Here, $\mu_k$ is a 30-dim vector with entries randomly selected from $\{-\delta_\mu, 0, \delta_\mu\}$.

Then, we train the student using

$$R_{\text{kd}}(f, \text{H}) \triangleq \frac{-1}{N} \sum_{n \in [N]} \left(P^{\text{tar}}_{x_n}\right)^T \cdot \log\left(P^s_{x_n}\right), \tag{20}$$

for 4 different values of $P^{\text{tar}}_{x_n}$: (i) Ground Truth, $P^{\text{tar}}_{x_n} = P^*_X$; (ii) MCMI KD, $P^{\text{tar}}_{x_n} = P^t_{x,\text{MCMI}}$; (iii) MLL KD, $P^{\text{tar}}_{x_n} = P^t_{x,\text{LL}}$; and (iv) On-Hot, $P^{\text{tar}}_{x_n} = \boldsymbol{e}(y)$.

In our experiments, we set $\sigma = 4$ and use different $\delta_\mu$ values as $\delta_\mu = \{0.5, 1, 2, 4\}$, and report the student's accuracy and the $L_2$ norm of the distance between $P^*_X$ and $P^{\text{tar}}_{x_n}$, denoted by Pdist in table 5 (the reason that we used $L_2$ norm, and not other distance metrics, is that the student's loss variance is upper-bounded by a function of $\|P^*_X - P^{\text{tar}}\|_2$ (Menon et al., 2021)).

As seen in table 5, compared to $P^t_{x,\text{LL}}$, $P^t_{x,\text{MCMI}}$ consistently provides a better estimate of $P^*_X$ leading to a higher test accuracy for the student.

Table 5: Results on synthetic Gaussian dataset.

| Dataset parameters | Metrics | Ground Truth | MCMI KD | MLL KD | One-Hot |
|---|---|---|---|---|---|
| $\delta_\mu = 0.5$ | Accuracy | $57.21 \pm 0.07$ | $57.02 \pm 0.18$ | $56.32 \pm 0.27$ | $55.48 \pm 0.55$ |
| | Pdist | $0$ | $13.25 \pm 0.04$ | $14.41 \pm 0.05$ | $41.23$ |
| $\delta_\mu = 1$ | Accuracy | $69.68 \pm 0.08$ | $69.32 \pm 0.09$ | $68.88 \pm 0.11$ | $67.76 \pm 0.24$ |
| | Pdist | $0$ | $9.02 \pm 0.07$ | $9.65 \pm 0.06$ | $34.52$ |
| $\delta_\mu = 2$ | Accuracy | $76.98 \pm 0.04$ | $76.54 \pm 0.06$ | $76.15 \pm 0.10$ | $75.71 \pm 0.15$ |
| | Pdist | $0$ | $4.42 \pm 0.02$ | $4.77 \pm 0.03$ | $7.11$ |
| $\delta_\mu = 4$ | Accuracy | $81.68 \pm 0.01$ | $81.60 \pm 0.02$ | $81.57 \pm 0.05$ | $81.44 \pm 0.08$ |
| | Pdist | $0$ | $2.44 \pm 0.02$ | $2.54 \pm 0.02$ | $3.18$ |

## A.7 POST-HOC SMOOTHING

It is known that by tuning the teacher's temperature $T^t$ and the student's temperature $T^s$, one can improve the student's accuracy (Liu et al., 2022). Here, we show that the improvements in the student's accuracy we obtained by using MCMI teachers cannot be obtained by such post-hoc smoothing method, i.e., changing the temperatures of the student and teacher's model.

To this end, we record the student's accuracy when trained by MLL and MCMI teachers for three different teacher-student pairs, where the teacher's temperature ranges in $T^t = \{1, 2, \ldots, 8\}$, and the student's temperature in $T^s = \{1, 2, 4\}$. Note that the $\lambda$ value for the MCMI teacher is the same for different combinations of $T^t$ and $T^s$, and it was not tuned for each combination.

The results are illustrated in fig. 8, where each row corresponds to a teacher-student pair. The observations are as follows:

As seen, no matter what combination of $T^t$ and $T^s$ is used for the student and teacher model, the student's accuracy is higher when using an MCMI teacher. Therefore, we conclude that, by using post-hoc smoothing method (temperature), the student's accuracy cannot improve as much as it can when using MCMI teacher.

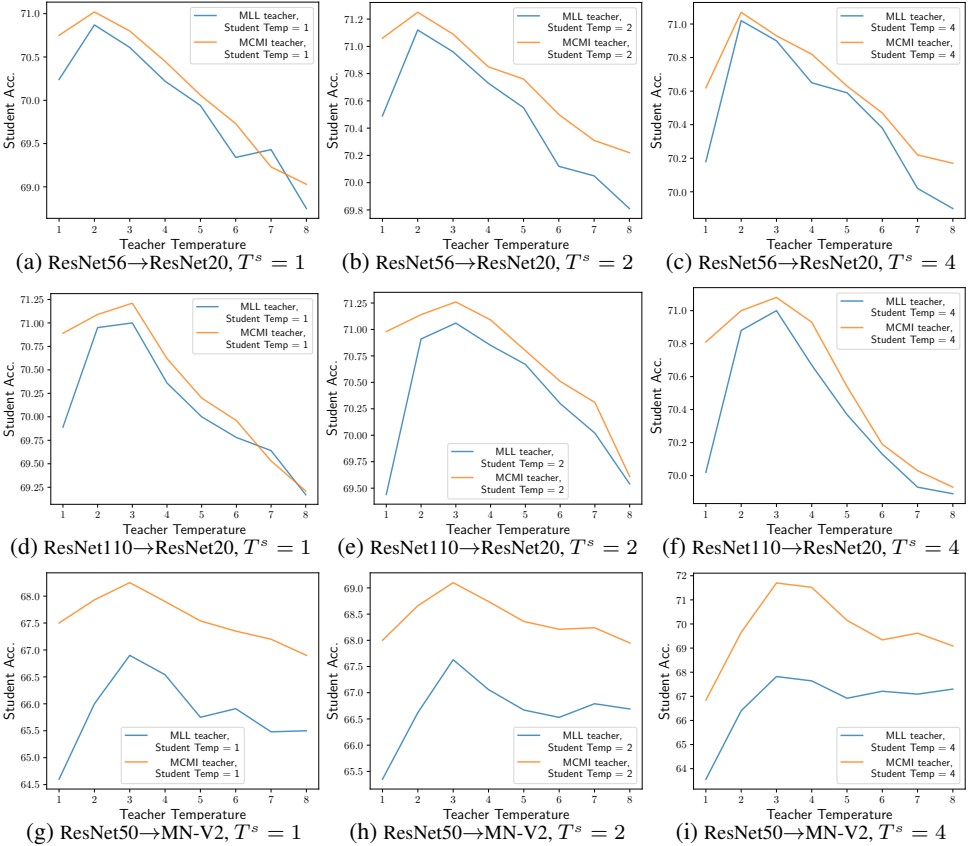

Figure 8: The student's accuracy in KD with various combinations of $T^t$ and $T^s$ values, when both the MCMI and MLL teachers are deployed. The teacher-student pairs are ResNet56-ResNet20 (top), ResNet110-ResNet20 (middle), ResNet50-MobileNetV2 (bottom).

## A.8 Implementation Details of the Experiments in section 6

### A.8.1 Fine-tuning setup for training MCMI teacher

As discussed, to train an MCMI teacher, we fine-tune a pre-trained teacher. For all the datasets, we obtain the pre-trained teacher from PyTorch official repository (Paszke et al., 2019) [6]. Then, for fine-tuning, we use cosine annealing learning rate scheduler with its initial value equals one-fifth of the initial learning rate that was used to train the pre-trained teacher. We utilized the same set of teacher weights across all experiments, wherein various KD variants were employed to train different student models. All the other training setups are the same as those used to train the pre-trained teacher, except the number of epochs for fine-tuning which is explained as follows:

For CIFAR-$\{10, 100\}$ we fine-tune the teachers with 20 epochs and test different $\lambda$ values, namely $\lambda = \{0.1, 0.15, 0.2, 0.25\}$[7], then we pick the best $\lambda$ value, denoted by $\lambda^*$. We list the $\lambda^*$ values for the teacher models on CIFAR-100 dataset in table 6.

Table 6: $\lambda^*$ values used for fine-tuning different teacher models on CIFAR-$\{10,100\}$ datasets.

|           | ResNet-56 | ResNet-110 | WRN-40-2 | VGG-13 | ResNet-50 | ResNet-32-4 |
|-----------|-----------|------------|----------|--------|-----------|-------------|
| CIFAR-100 | 0.15      | 0.10       | 0.15     | 0.20   | 0.20      | 0.15        |
| CIFAR-10  | 0.25      | -          | -        | -      | -         | -           |

For ImageNet, we use 10 epochs to fine-tune the pre-trained teachers, and test different $\lambda$ values, namely $\lambda = \{0.05, 0.1, 0.15, 0.2, 0.3\}$, Specifically, we pick $\lambda^* = 0.1$ and $\lambda^* = 0.15$ for ResNet-34 and ResNet-50, respectively.

In fact, we opted to fine-tune the model for 20 and 10 epochs for CIFAR and ImageNet, respectively, due to two primary reasons.

1. Tuning the trade-off: We can effectively establish the desired trade-off between LL and CMI values by adjusting the value of $\lambda$, which serves as a weight parameter. Going beyond 20 (10) epochs on CIAFR (ImageNet) datasets does not offer significant benefits in achieving this trade-off, making it a practical and efficient choice for model tuning.

2. Time complexity consideration: We aim to keep the number of fine-tuning epochs relatively low to avoid introducing excessive time complexity into the knowledge distillation (KD) process. This ensures that the overall training process remains efficient.

### A.8.2 knowledge distillation algorithm setup

For all variants of knowledge distillation and settings in this paper, SGD is applied as the optimizer.

For CIFAR-100 dataset, we train 240 epochs for all experiments with an initial learning rate of 0.05 by default, which will be decayed by factor of 0.1 at epoch 150, 180, 210. For MobileNetV2, ShuffleNetV1, and ShuffleNetV2. a smaller initial learning rate of 0.01 is used. We adopt batch size of 64.

For ImageNet dataset, we used the same training recipe as PyTorch official implementation, except train for 10 more epochs. Batch size of 256 is adopted.

In the paper, we incorporate several state-of-the-art KD variants. Specifically, we employ a linear combination of cross-entropy loss and knowledge distillation loss, as outlined below:

$$\ell = \ell_{\text{CE}} + \gamma \ell_{\text{Distill}} \tag{21}$$

For different KD variants, we directly adopted the $\gamma$ values as those reported in their original papers. The $\gamma$ values and additional details for each KD variant used in the paper are outlined below:

1. AT (Zagoruyko & Komodakis, 2016): $\gamma = 1000$;

---

[6]https://pytorch.org/vision/stable/models.html.

[7]In table 7,we study how different $\lambda$ values affect the teacher's CMI and LL, and the student's accuracy.

2. PKT (Passalis & Tefas, 2018): $\gamma = 30000$;

3. SP (Tung & Mori, 2019): $\gamma = 3000$;

4. CC (Peng et al., 2019): $\gamma = 0.02$;

5. RKD (Park et al., 2019a): $\gamma_1 = 25$ for distance, and $\gamma_2 = 50$ for angle;

6. VID (Ahn et al., 2019): $\gamma = 1$;

7. CRD (Tian et al., 2019): $\gamma = 0.8$;

8. DKD (Zhao et al., 2022): Consistent with their official implementation, we select $\gamma$ from the set $\{0.2, 1, 2, 4, 8\}$;

9. REVIEWKD Chen et al. (2021a): Consistent with their official implementation, we select $\gamma$ from the set $\{0.6, 1, 5, 8\}$;

10. HSAKD (Yang et al., 2021): $\gamma = 1$.

For KD, we follow the original implementation in (Hinton et al., 2015), set $\alpha = 0.9$, and $T = 4$.

For both zero-shot and few-shot, we follow the same training recipe as that discussed in this subsection.

### A.8.3 Effect of $\lambda$ in MCMI

In this subsection, our objective is to investigate the impact of the parameter $\lambda$ in MCMI estimation when maximizing the objective function in eq. (14). To this end, we fine-tune a pre-trained teacher employing $\lambda = \{0, 0.005, 0.01, 0.1, 0.15, 0.2, 0.25\}$. As expected, as $\lambda$ increases, the teacher's CMI value also increases while its LL value decreases. In addition, the student's accuracy forms a quasiconcave-like function w.r.t. $\lambda$, and it reaches its maximum value at $\lambda^* = 0.15$. In fact, at this point, a good trade-off between the teacher's CMI and LL values is established yielding a good BCPD estimate by the teacher.

Table 7: The effect of $\lambda$ on student's accuracy and Teacher's CMI and LL values. The teacher-student pair is ResNet-56→ResNet-20, and all the values are averaged over 5 different runs.

| $\lambda$ | 0 | 0.005 | 0.01 | 0.05 | 0.1 | 0.15 | 0.2 | 0.25 |
|---|---|---|---|---|---|---|---|---|
| Student's Acc. | 70.62 | 70.67 | 70.73 | 70.78 | 70.81 | 70.84 | 70.73 | 70.66 |
| Teacher's CMI | 0.1602 | 0.3045 | 0.3830 | 0.4053 | 0.4326 | 0.4428 | 0.4430 | 0.4424 |
| Teacher's LL | -0.114 | -0.212 | -0.317 | -0.325 | -0.336 | -0.344 | -0.352 | -0.363 |

## A.9    EFFECTIVENESS OF MCMI TEACHER IN SEMI-SUPERVISED DISTILLATION

Semi-supervised learning (Chen et al., 2020b; Iliopoulos et al., 2022) is a popular technique due to its ability to generate pseudo-labels for larger unlabeled dataset. In the context of KD, the teacher is indeed responsible for generating pseudo-labels for new, unlabeled examples.

In order to evaluate the MCMI teacher's performance under semi-supervised learning scenario, we conducted experiments on CIFAR-10 dataset, follow the setting of Chen et al. (2020b), with ResNet18 as the student model. The results we present below are averaged over 3 runs. In this setup, although all the 50000 training images are available, only a small percentage of them are labeled—1% (500), 2% (1000), 4% (2000), and 8% (4000)—. The results are depicted in appendix A.9, where the student accuracy is plotted as a function of number of labeled samples. As seen, not only can the MCMI teacher effectively perform in the semi-supervised distillation scenario, but it also outperforms the MLL teacher.

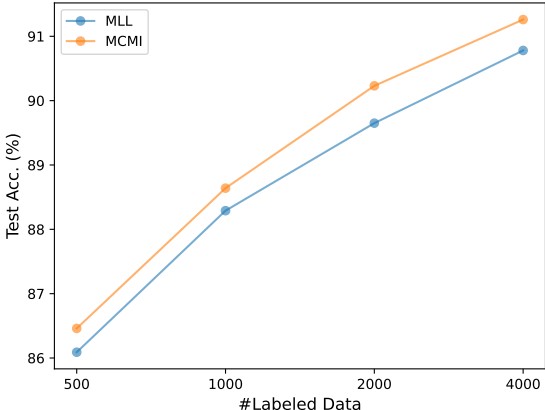

Figure 9: Effectiveness of MCMI teacher in semi-supervised distillation.

## A.10 EIGEN-CAM FIGURES

In this section, we visualize the activation maps of MCMI and MLL teacher for 4 different classes of ImageNet datasets, namely {goldfish, mountain tent, toy terrier, bee eater}. To achieve this, we have randomly chosen 24 samples from each of these four classes. In addition, we compare the CMI value of MCMI and MLL teacher over these four classes in table 8.

Focusing on Eigen-CAM figures for a specific class, let's say "mountain tent", we observe that the MCMI teacher activation maps have more randomness compared to the MLL teacher. To shed more light, the MLL teacher always highlights the tent itself and this is why it has a higher accuracy compare to the MCMI teacher. On the other hand, the MCMI teacher is more capable of capturing the contextual information as it sometimes highlight the tent as a whole, sometimes the details of the tent, and sometimes some other objects in the background of the image.

Table 8: Comparing CMI values between the MLL and MCMI teachers across four distinct classes from the ImageNet dataset.

| Classes | goldfish | mountain tent | toy terrier | bee eater |
|---|---|---|---|---|
| MLL | 0.2896 | 1.1694 | 0.4132 | 0.2954 |
| MCMI | 0.4173 | 1.2312 | 0.5414 | 0.3471 |

### A.10.1 CLASS "MOUNTAIN TENT"

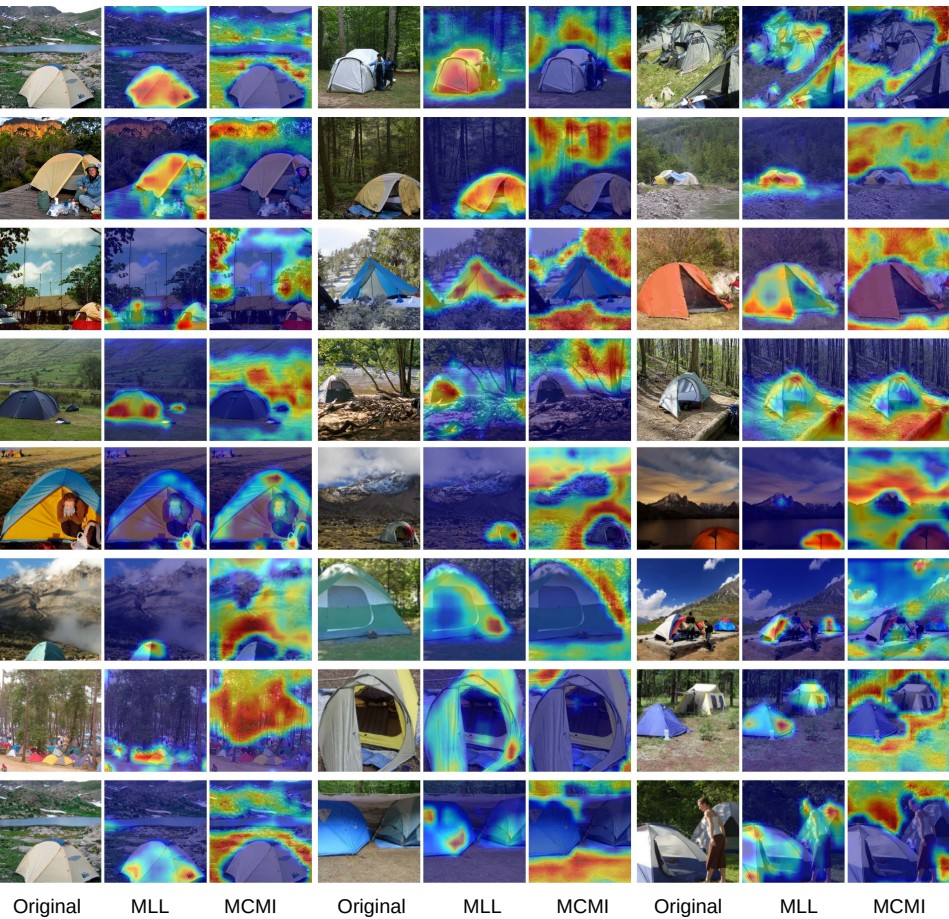

| Original | MLL | MCMI | Original | MLL | MCMI | Original | MLL | MCMI |

Figure 10: Eigen-CAM for 24 randomly selected samples from class "tent" in ImageNet, for MLL and MCMI teachers.

## A.10.2 Class "goldfish"

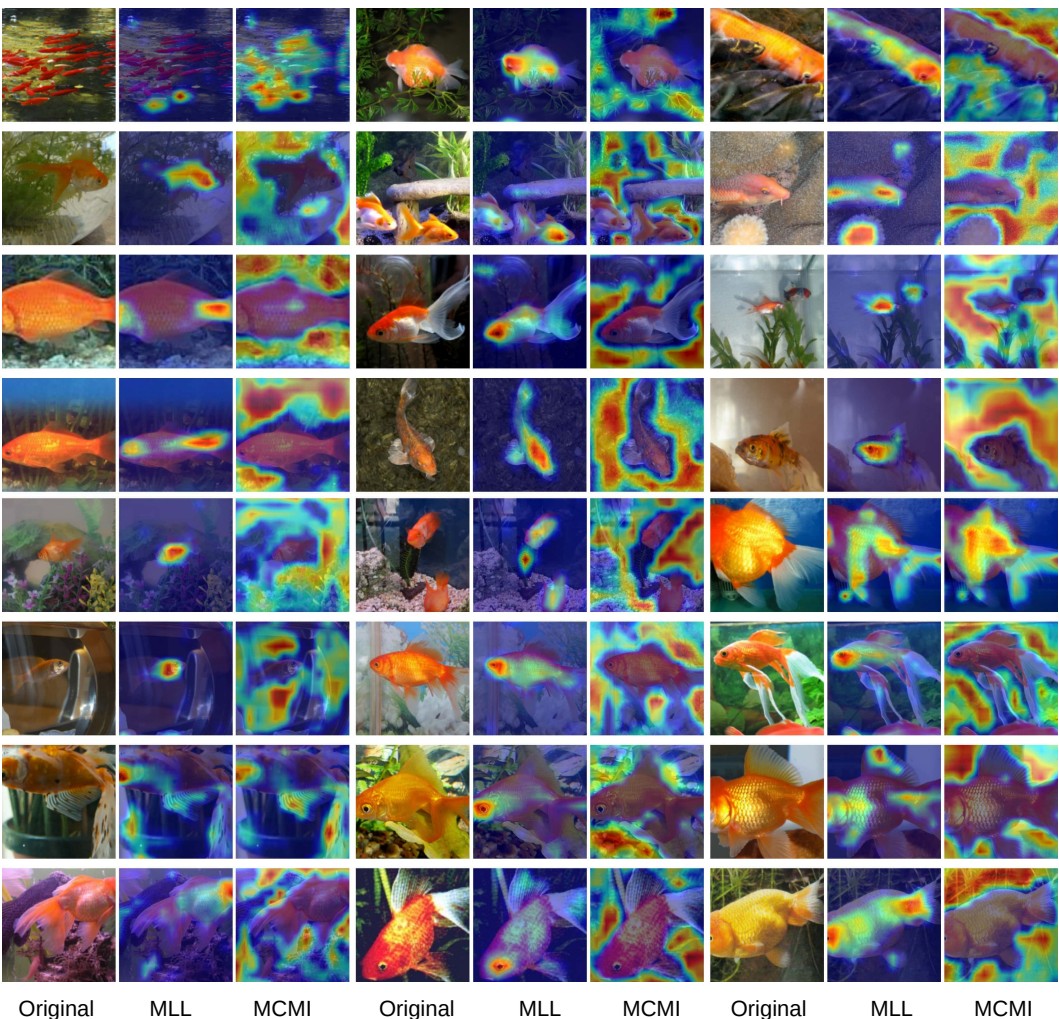

Original    MLL    MCMI     Original    MLL    MCMI     Original    MLL    MCMI

Figure 11: Eigen-CAM for 24 randomly selected samples from class "goldfish" in ImageNet, for MLL and MCMI teachers.

### A.10.3 CLASS "BEE EATER"

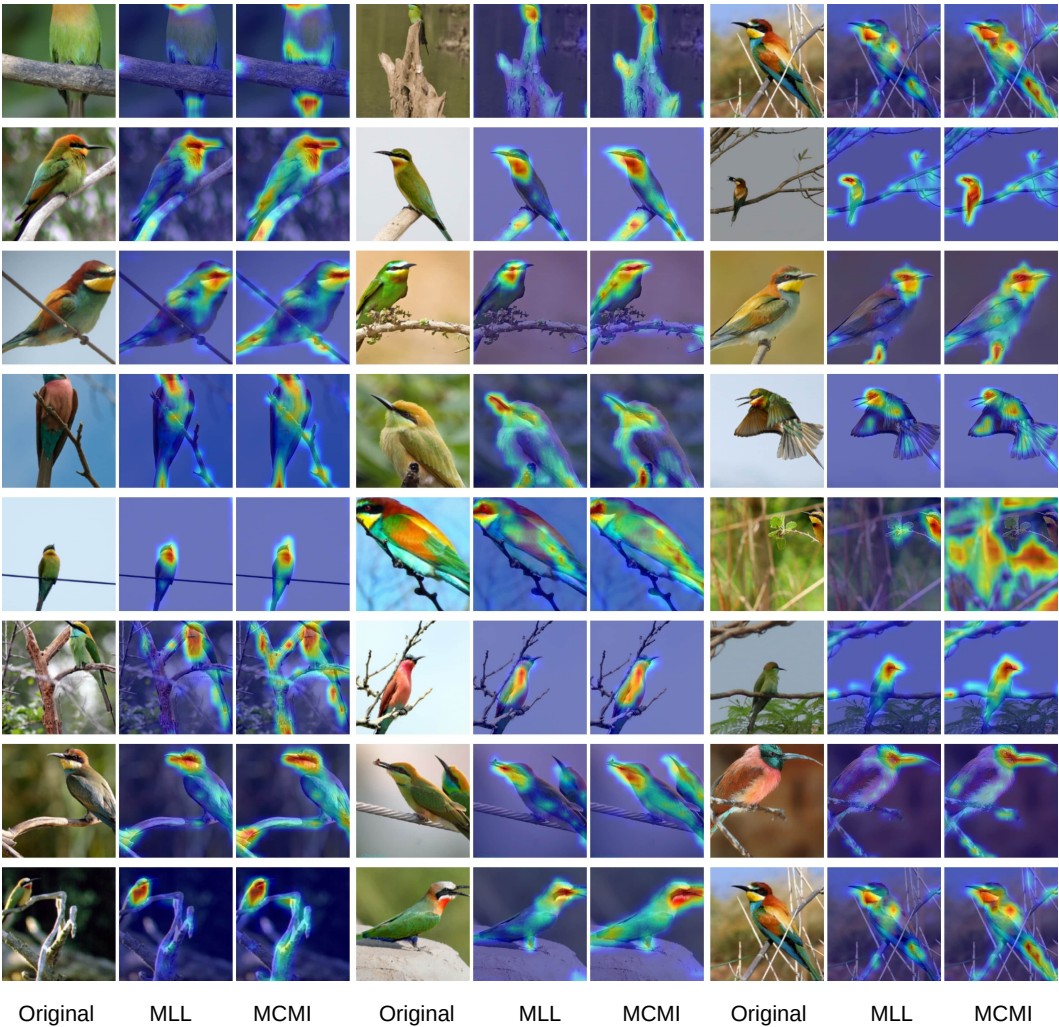

Original  MLL  MCMI  Original  MLL  MCMI  Original  MLL  MCMI

Figure 12: Eigen-CAM for 24 randomly selected samples from class "bee eater" in ImageNet, for MLL and MCMI teachers.

### A.10.4 CLASS "TOY TERRIER"

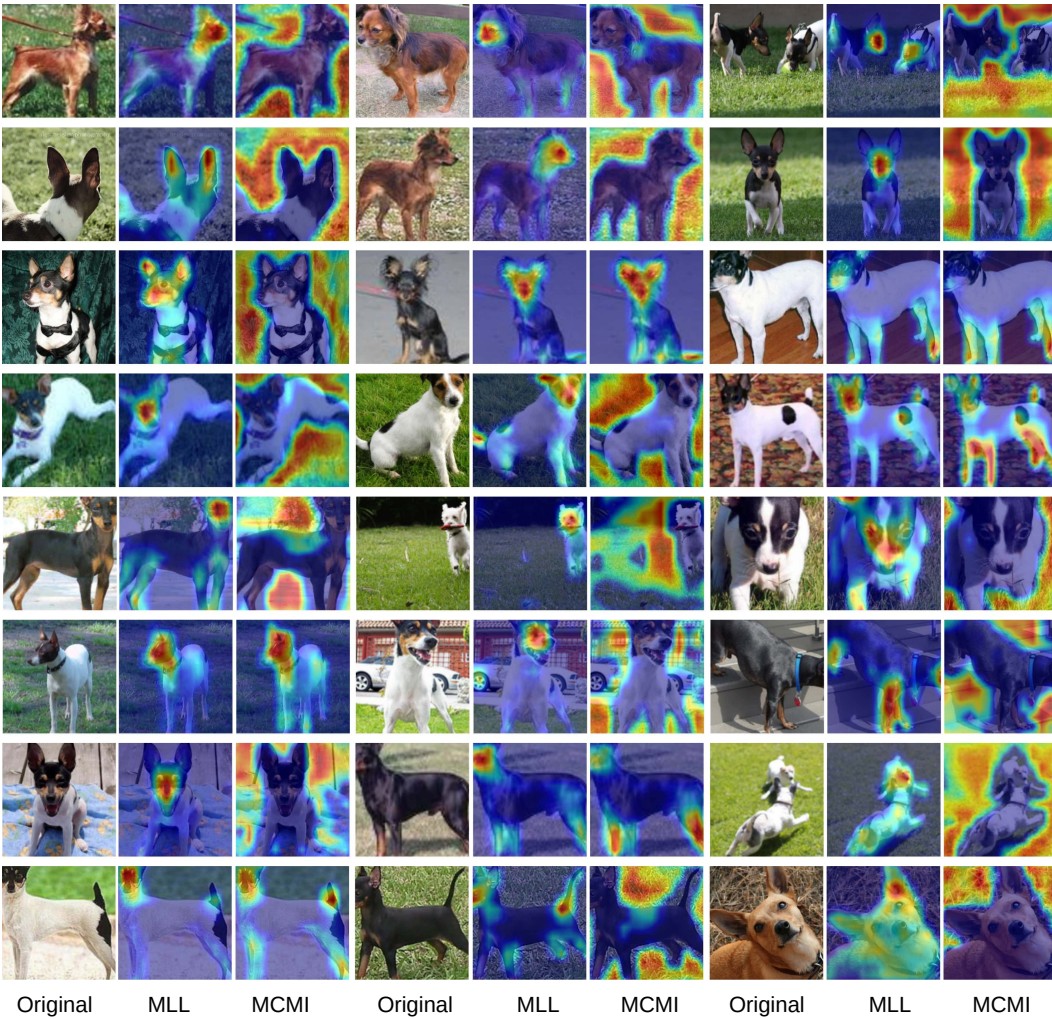

| Original | MLL | MCMI | Original | MLL | MCMI | Original | MLL | MCMI |

Figure 13: Eigen-CAM for 24 randomly selected samples from class "Toy Terrier" in ImageNet, for MLL and MCMI teachers.

### A.11 Zero-shot KD

#### A.11.1 Real-world examples for zero-shot KD

Zero-shot learning in the context of knowledge distillation (KD) can be used when you want a student model to learn from a teacher model while omitting certain classes. In this subsection, we will provide detailed explanations of two real-world examples that show the application of zero-shot learning within the framework of KD.

Consider the scenario that you are tasked with developing a disease classification system using medical images as input data. The central repository of these medical images is managed by the government, which possesses the teacher model designed for this purpose. However, local hospitals express interest in utilizing these images for disease diagnosis. Due to privacy concerns, sharing all the images with these hospitals is not feasible McMahan et al. (2017); Hamidi & Yang (2024); Guo et al. (2021); Hamidi & Damen (2024); Hamidi et al. (2022).

Here's how zero-shot learning in KD with omitted classes could be applied:

1. Teacher Model: You have a teacher model that has been trained on a large medical images including those sensitive ones.

2. Omitted Classes: Your objective is to create a student model for disease diagnosis. However, you want to exclude certain sensitive images from the training dataset.

3. Knowledge Distillation: You create a student model for disease classification and initiate knowledge distillation from the teacher model.

4. Zero-Shot Learning: The student model, can still perform diagnosing effectively, even for sensitive images that were omitted during training. It achieves this through zero-shot learning, as it learns to predict diseases that have never seen directly in the training data by leveraging the knowledge from the teacher model.

In this scenario, zero-shot learning in KD allows you to build a classification system that respects **privacy** and **confidentiality** by omitting certain classes (sensitive images) while ensuring that the diagnosing quality remains high for other deceases.

#### A.11.2 Comparing to conventional zero-shot learning

As discussed in the main body of the paper, the MCMI teacher can help the student to perform *generalized* zero-shot learning, where the task is to classify samples from both seen and unseen classes.

The methods in the literature proposed for *generalized* zero-shot learning are based on the assumption that we have access to the class semantic vectors that provide descriptions of classes (Akata et al., 2015; Huynh & Elhamifar, 2020; Xian et al., 2017; Bucher et al., 2017). A common approach to use such semantics for classifying samples from both seen and unseen classes is to employ positive/negative scoring, which assigns positive scores to related semantics and negative scores to unrelated semantics.

This method has a major downfall: the semantic vectors for both seen and unseen classes should be available at the training time. This will, in turn, hinder applicability of these methods. However, our method for *generalized* zero-shot learning does not depend on such information. We further note that our method is generic in that it does not matter which class(es) is (are) missing in the student's training dataset.

#### A.11.3 Why our method works for zero-shot learning?

Here, we clarify the reason why our method can classify the missing classes. This is due to the fact that the information regarding the labels for the missing classes is still contained within $p_x^*$ of the samples presented in the dataset. To elucidate, consider the following example in which the objective is to train a DNN to classify three classes: dog, cat, and car. Assume that the training samples for dog is entirely missing from the training dataset. Suppose that for a particular training sample from the cat class, $p_x^* = [0.30, 0.68, 0.02]$, with the probability values corresponding to the dog, cat, and car classes, respectively. The value of 0.30 suggests that this sample exhibits some features resembling

the dog class, such as a similarity in the shape of legs. Such information provided from the BCPD vectors for the samples of existing classes can collectively help the DNN to classify the samples for the missing class. We note that the one-hot vector for this sample is $[0, 1, 0]$, and therefore, such information does not exist when one-hot vectors are used as unbiased estimate of $p_x^*$.

### A.11.4  MORE ZERO-SHOT KD RESULTS

• **CIFAR-100:** We consider the zero-shot in KD over CIFAR-100 where the teacher is trained over the whole dataset while some number of classes are dropped during the distillation. Specifically, we consider dropping $\{5, 10, 20, 30, 40, 50\}$ classes. The teacher and student models are WRN-40-2 and WRN-16-2, respectively. We compare using the MLL and MCMI teacher in the above scenario. Then, we report the accuracy of the student for both dropped and preserved classes, separately.

The results are summarized in table 9. As the results suggest, the accuracy for the dropped classes is significantly increases by using the MCMI teacher. In addition, the accuracy for the preserved classes is almost the same when using either teacher. This suggests that the accuracy for the preserved classes is not sacrificed for a higher accuracy for the dropped classes.

Table 9: Student accuracy of zero-shot learning in CIFAR-100, where teacher is WRN-40-2, and student is WRN-16-2. Numbers in red indicate the accuracy of omitted classes when training students, those in blue represent that of preserved classes. The reported results are averaged over 5 different random seeds.

| # of dropped classes | 5 | 10 | 20 | 30 | 40 | 50 |
|---|---|---|---|---|---|---|
| MLL | 0.16 / 75.02 | 2.70 / 75.82 | 1.63 / 76.28 | 1.11 / 80.15 | 1.74 / 80.42 | 0.87 / 82.38 |
| MCMI | 25.64 / 75.08 | 32.78 / 75.78 | 30.61 / 76.22 | 21.72 / 80.27 | 23.74 / 80.79 | 22.26 / 82.46 |

• **CIFAR-10:** For this dataset, we use a $10 \times 10$ confusion matrix $\mathbf{C}$ to better observe the model performance. Each row in $\mathbf{C}$ represents the instances in an actual class, while each column represents the instances predicted by the model. In particular, the entry $\mathbf{C}_{i,i}$, $i \in [10]$, represents the probability that the instances in class $i$ are correctly classified. On the other hand, $\mathbf{C}_{i,j}$, $i, j \in [10], i \neq j$, represents the probability of the cases where instances from class $i$ are incorrectly classified as belonging to Class $j$.

Then, we consider dropping $\{1, 2, 3, 4, 5, 6\}$ classes from the student's training set. The dropped classes are denoted by red color in the confusion matrices.

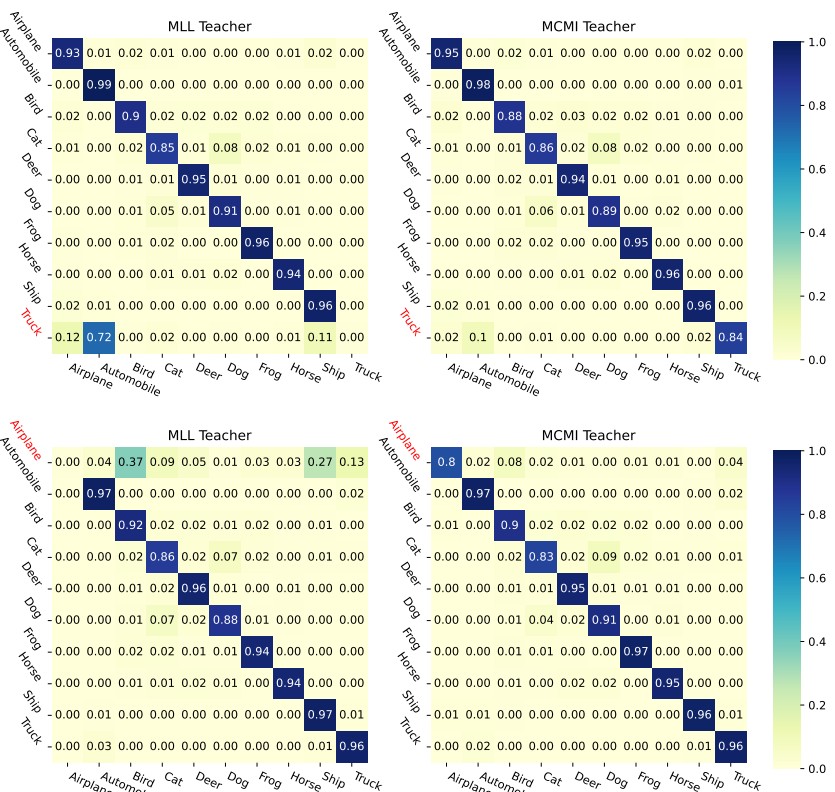

Figure 14: Zero-shot KD when one of the classes is entirely removed for the student.

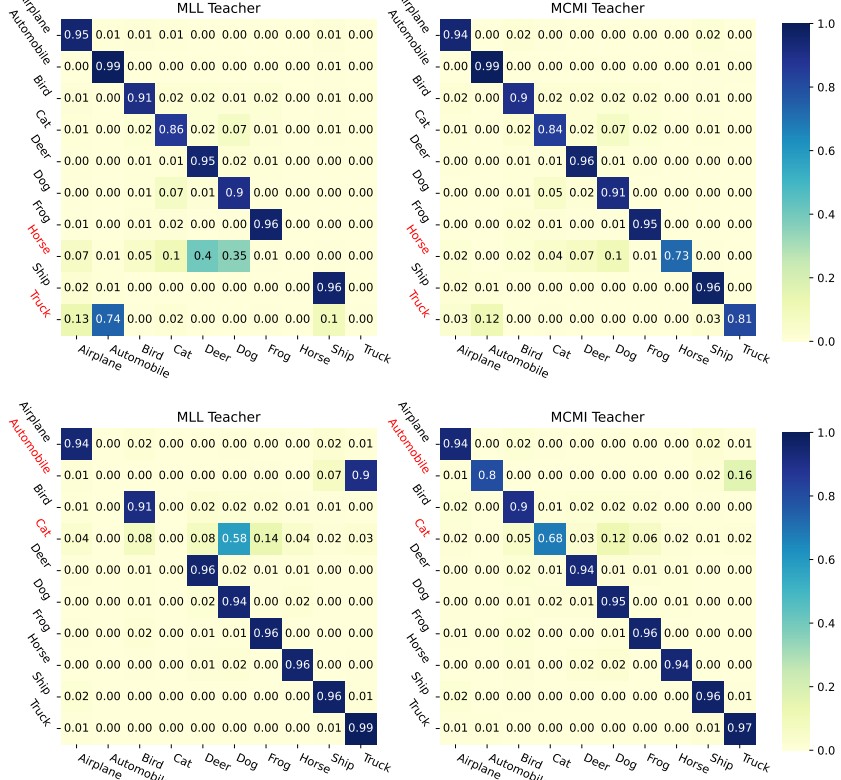

Figure 15: Zero-shot KD when two classes are entirely removed for the student.

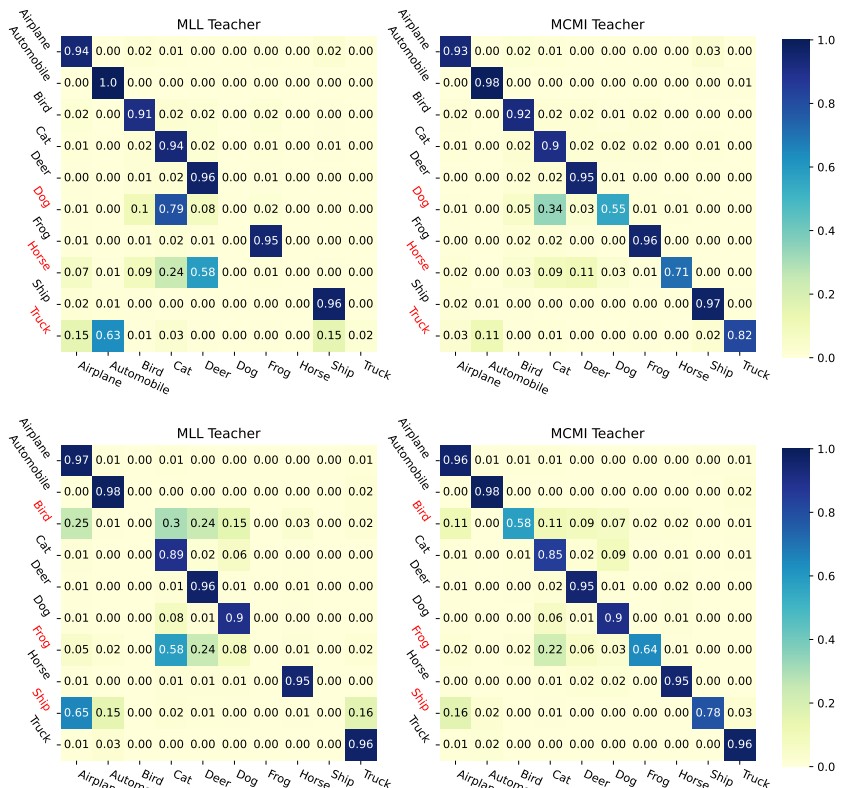

Figure 16: Zero-shot KD when three classes are entirely removed for the student.

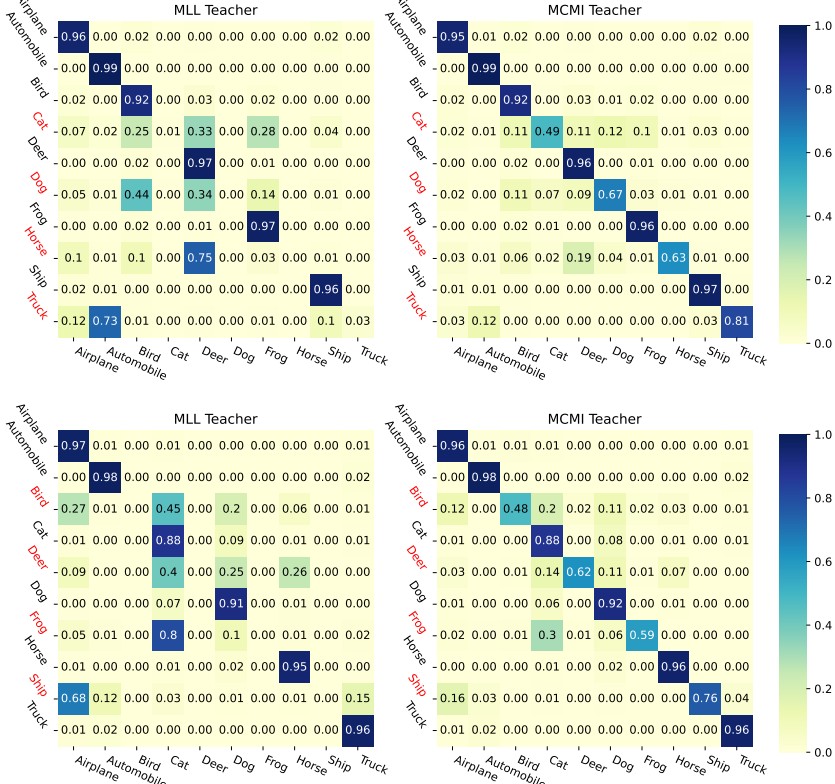

Figure 17: Zero-shot KD when four classes are entirely removed for the student.

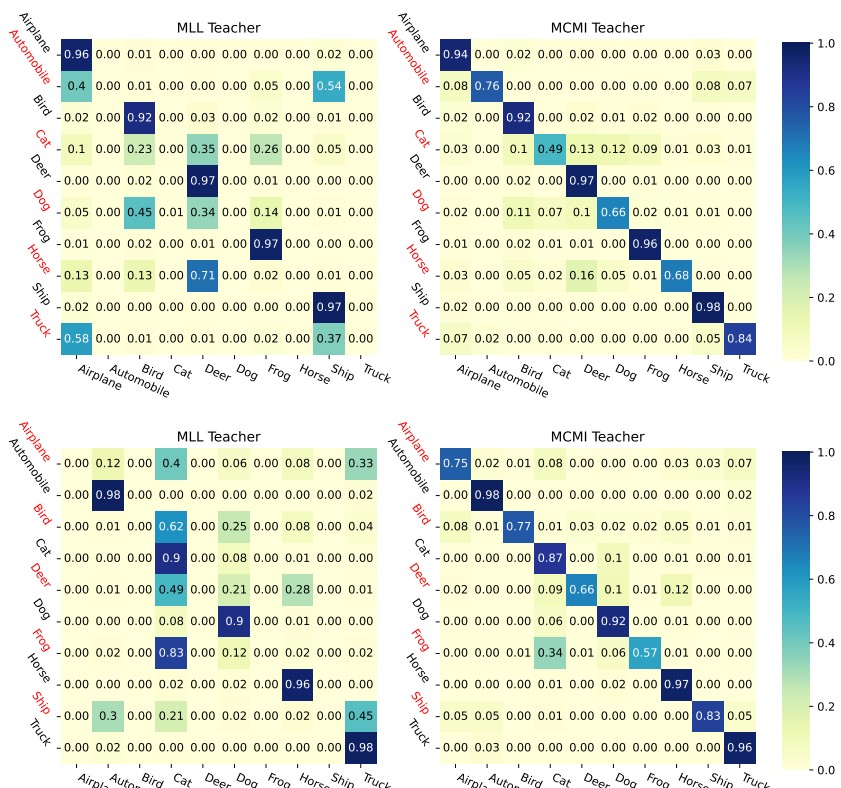

Figure 18: Zero-shot KD when five classes are entirely removed for the student.

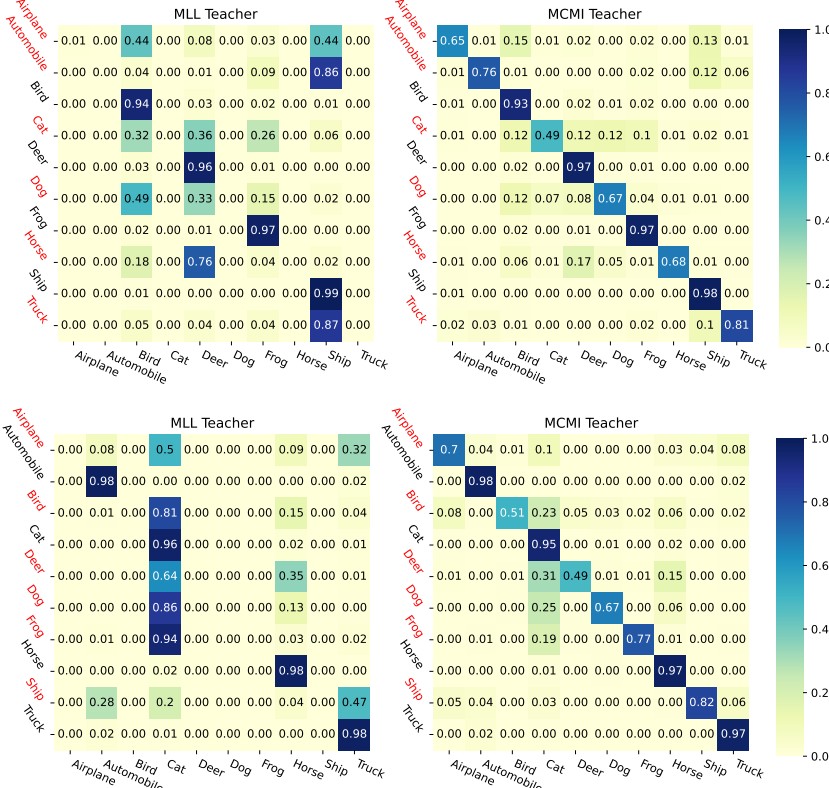

Figure 19: Zero-shot KD when six classes are entirely removed for the student.

## A.12 ACCURACY VARIANCES

Table 10: The variance in accuracy for the results in Table 1, where the teacher and student have the **same** architectures.

| Teacher | ResNet-56 | ResNet-110 | ResNet-110 | WRN-40-2 | WRN-40-2 | VGG-13 |
| Student | ResNet-20 | ResNet-20 | ResNet-32 | WRN-16-2 | WRN-40-1 | VGG-8 |
|---|---|---|---|---|---|---|
| KD | $70.84 \pm 0.10$ | $70.85 \pm 0.20$ | $73.48 \pm 0.13$ | $75.42 \pm 0.12$ | $74.53 \pm 0.34$ | $73.83 \pm 0.22$ |
| AT | $70.89 \pm 0.28$ | $70.68 \pm 0.32$ | $73.96 \pm 0.18$ | $74.49 \pm 0.15$ | $73.25 \pm 0.28$ | $71.76 \pm 0.16$ |
| PKT | $70.96 \pm 0.38$ | $71.03 \pm 0.12$ | $72.92 \pm 0.14$ | $75.01 \pm 0.15$ | $74.15 \pm 0.28$ | $73.35 \pm 0.20$ |
| SP | $70.98 \pm 0.07$ | $70.83 \pm 0.26$ | $73.34 \pm 0.18$ | $74.60 \pm 0.27$ | $73.60 \pm 0.16$ | $73.29 \pm 0.14$ |
| CC | $69.98 \pm 0.47$ | $70.02 \pm 0.18$ | $71.71 \pm 0.23$ | $74.00 \pm 0.24$ | $72.50 \pm 0.16$ | $71.02 \pm 0.29$ |
| RKD | $70.68 \pm 0.13$ | $70.24 \pm 0.17$ | $72.65 \pm 0.24$ | $73.97 \pm 0.15$ | $72.66 \pm 0.21$ | $72.03 \pm 0.28$ |
| VID | $70.64 \pm 0.25$ | $70.69 \pm 0.09$ | $73.10 \pm 0.13$ | $74.44 \pm 0.25$ | $73.58 \pm 0.28$ | $71.93 \pm 0.10$ |
| CRD | $71.40 \pm 0.13$ | $71.93 \pm 0.17$ | $74.03 \pm 0.08$ | $75.82 \pm 0.15$ | $74.86 \pm 0.11$ | $74.23 \pm 0.05$ |
| DKD | $72.31 \pm 0.12$ | $71.83 \pm 0.18$ | $74.36 \pm 0.14$ | $76.66 \pm 0.22$ | $75.63 \pm 0.09$ | $74.87 \pm 0.11$ |
| REVIEWKD | $72.31 \pm 0.26$ | $72.11 \pm 0.30$ | $74.01 \pm 0.43$ | $76.29 \pm 0.16$ | $75.47 \pm 0.14$ | $74.96 \pm 0.18$ |
| HSAKD | $72.70 \pm 0.28$ | $73.15 \pm 0.16$ | $75.71 \pm 0.13$ | $77.36 \pm 0.17$ | $77.55 \pm 0.22$ | $75.86 \pm 0.20$ |

Table 11: The variance in accuracy for the results in Table 1, where the teacher and student have **different** architectures.

| Teacher | ResNet-50 | ResNet-50 | ResNet-32×4 | ResNet-32×4 | WRN-40-2 | VGG-13 |
| Student | MobileNetV2 | VGG-8 | ShuffleNetV1 | ShuffleNetV2 | ShuffleNetV1 | MobileNetV2 |
|---|---|---|---|---|---|---|
| KD | $70.23 \pm 0.34$ | $74.59 \pm 0.16$ | $75.90 \pm 0.11$ | $76.32 \pm 0.47$ | $76.45 \pm 0.27$ | $69.14 \pm 0.22$ |
| AT | $60.03 \pm 0.21$ | $72.19 \pm 0.94$ | $75.05 \pm 0.17$ | $75.21 \pm 0.13$ | $75.61 \pm 0.25$ | $62.07 \pm 0.26$ |
| PKT | $67.42 \pm 0.18$ | $73.43 \pm 0.17$ | $75.21 \pm 0.27$ | $76.34 \pm 0.18$ | $75.39 \pm 0.14$ | $68.37 \pm 0.35$ |
| SP | $69.07 \pm 0.17$ | $74.14 \pm 0.19$ | $76.56 \pm 0.29$ | $76.70 \pm 0.18$ | $76.82 \pm 0.14$ | $67.83 \pm 0.28$ |
| CC | $66.76 \pm 0.42$ | $70.90 \pm 0.20$ | $71.77 \pm 0.17$ | $73.02 \pm 0.17$ | $71.80 \pm 0.19$ | $65.45 \pm 0.39$ |
| RKD | $65.11 \pm 0.49$ | $72.10 \pm 0.27$ | $73.59 \pm 0.18$ | $74.67 \pm 0.29$ | $74.26 \pm 0.31$ | $65.37 \pm 0.10$ |
| VID | $67.61 \pm 0.19$ | $70.69 \pm 0.17$ | $74.58 \pm 0.23$ | $74.67 \pm 0.25$ | $75.03 \pm 0.17$ | $65.77 \pm 0.36$ |
| CRD | $69.70 \pm 0.69$ | $74.86 \pm 0.17$ | $76.82 \pm 0.22$ | $77.54 \pm 0.10$ | $76.62 \pm 0.27$ | $69.98 \pm 0.13$ |
| DKD | $71.70 \pm 0.35$ | $75.35 \pm 0.13$ | $77.21 \pm 0.14$ | $77.66 \pm 0.19$ | $77.42 \pm 0.45$ | $70.35 \pm 0.08$ |
| REVIEWKD | $70.63 \pm 0.26$ | $74.20 \pm 0.85$ | $77.78 \pm 0.31$ | $78.23 \pm 0.14$ | $77.56 \pm 0.32$ | $71.70 \pm 0.46$ |
| HSAKD | $72.98 \pm 0.26$ | $76.45 \pm 0.19$ | $79.77 \pm 0.09$ | $80.01 \pm 0.16$ | $78.87 \pm 0.42$ | $72.80 \pm 0.11$ |

Table 12: The variance in accuracy of few shot experiments in Table 3.

| Percentage | 5% | | 10% | | 15% | | 25% | | 35% | | 50% | | 75% | |
| Teacher | MLL | MCMI | MLL | MCMI | MLL | MCMI | MLL | MCMI | MLL | MCMI | MLL | MCMI | MLL | MCMI |
|---|---|---|---|---|---|---|---|---|---|---|---|---|---|---|
| KD | 52.30 | 58.02 | 60.13 | 63.75 | 63.52 | 66.20 | 66.78 | 68.10 | 68.28 | 69.34 | 69.52 | 70.28 | 70.44 | 70.59 |
| | ±0.26 | ±0.21 | ±0.29 | ±0.17 | ±0.32 | ±0.23 | ±0.29 | ±0.68 | ±0.37 | 0.32 | 0.22 | ±0.23 | ±0.23 | ±0.46 |
| CRD | 47.60 | 51.40 | 54.60 | 56.80 | 58.90 | 60.02 | 63.82 | 64.70 | 66.70 | 67.22 | 68.84 | 69.15 | 70.35 | 70.40 |
| | ±0.18 | ±0.19 | ±0.41 | ±0.11 | ±0.22 | ±0.29 | ±0.19 | ±0.77 | ±0.23 | ±0.20 | ±0.24 | ±0.76 | ±0.26 | ±0.88 |

## A.13 BINARY CLASSIFICATION ON CUSTOMIZED CIFAR-$\{10, 100\}$

It is known that the improvement in student's accuracy in binary classification is typically lower than that observed in multi-class classification scenarios since the amount of information transferred from the teacher to the student network is restricted (Sajedi & Plataniotis, 2021; Tzelepi et al., 2021a;b; Müller et al., 2020).

In this section, we want to empirically verify the effectiveness of MCMI teacher in binary classification tasks. To this end, we create three binary classification datasets from CIFAR-$\{10, 100\}$ datasets as explained in the sequel.

• **Dataset 1:** Similarly to (Müller et al., 2020), we create $CIFAR - 2 \times 5$ dataset, where the input distribution has "sub-classes"- structure, i.e., we merge the first 5 classes of CIFAR-10 as class one, and the remaining 5 classes as class two.

• **Dataset 2:** We keep the training/testing samples from only two classes in the CIFAR-100 dataset: those belonging to class 26 (crab) and class 45 (lobster), creating a dataset referred to as CIFAR-26-45.

• **Dataset 3:** Similar to dataset 2, but we keep class 50 (mouse) and 74 (shrew) from CIFAR-100, which we refer to as CIFAR-50-74.

Now, we use VGG-13 and MobileNetv2-0.1 as the teacher and student models, respectively; and use conventional KD, AT, CC and VID to distill knowledge from MLL and MCMI teachers to the student.

The results for all the three tasks are listed in Table 13, with the values averaged over five different runs. As seen, the MCMI teacher yields a better student's accuracy in most of the cases. Also, it is worth noting that in some cases, for instance, AT over $CIFAR - 2 \times 5$ dataset, the distillation method hurts the accuracy of the student.

Table 13: Results of binary classification knowledge distillation on variants of CIFAR dataset.

| Dataset | $CIFAR - 2 \times 5$ | | CIFAR-26-45 | | CIFAR-50-74 | |
|---|---|---|---|---|---|---|
| Student Acc. | 76.94 | | 65.50 | | 66.30 | |
| Teacher | MLL | MCMI | MLL | MCMI | MLL | MCMI |
| KD | 77.00 | 77.41 (+0.41) | 69.70 | 70.10 (+0.40) | 71.40 | 71.20 (-0.20) |
| AT | 67.39 | 68.19 (+0.80) | 64.70 | 65.43 (+0.73) | 67.10 | 69.50 (+2.40) |
| CC | 77.19 | 77.85 (+0.66) | 70.30 | 70.75 (+0.45) | 69.70 | 71.40 (+1.70) |
| VID | 77.16 | 77.34 (+0.18) | 69.25 | 70.00 (+0.75) | 69.80 | 69.80 (+0.00) |

