# OpenReview forum: "Bayes Conditional Distribution Estimation for Knowledge Distillation Based on Conditional Mutual Information"
_ICLR.cc/2024/Conference — ICLR 2024 poster_

### Official Review · Reviewer_AG5J · 2023-10-29

**Soundness:** 4 excellent
**Presentation:** 4 excellent
**Contribution:** 3 good
**Rating:** 8
**Confidence:** 4

**Summary:**

This paper proposed a new distillation technique which is based on training teacher models so that they are well-suited for conveying information to the student models. Towards that end, the authors introduce a"conditional mutual information"(CMI) objective into the training process of the teacher model, whose goal is to improve the teacher's Bayes conditional probability estimates (via its soft-labels) — according to recent knowledge-distillation literature, more accurate Bayes conditional probability estimates result in better student's performance.

Overall:

(i) The authors argue that the so-called dark knowledge passed by the teacher to the student is the contextual information of the images which can be quantified via the conditional mutual information.
(ii) They provide evidence that temperature-scaling in KD increases the teacher's CMI value
(iii) They provide evidence that show that models with lower CMI values are not good teacher's, even if they're more accurate.
(iv) They provide experiments on CIFAR-100 and Imagenet datasets showing evidence that their method helps in improving the student's performance, compared to other standard distillation techniques.
(v) They show that their technique is especially effective in few-shot and zero-shot settings.

**Strengths:**

This is a well-written paper that presents a novel approach to knowledge distillation. They authors have provided extensive experimental evidence.

**Weaknesses:**

— The role of the teacher as a "provider of estimates for the unknown Bayes conditional probability distribution" is a theory for why distillation works that applies well mainly in the context of multi-class classification, and especially in the case where the input is images. (Indeed, there are other explanations for why knowledge distillation works, as it can be seen as a curriculum learning mechanism, a regularization mechanism etc see e.g. [1])

In that sense, I feel that the author should either make the above more explicit in the text, i.e., explicitly restrict the scope of their claims to multi-classifcation and images, or provide evidence that their technique gives substantial improvements on binary classification tasks in NLP datasets (but even in vision datasets).

— One of the main reasons why knowledge distillation is such a popular technique, is because the teacher can generate pseudo-labels for new, unlabeled examples, increasing the size of the student's dataset. (This is known as semi-supervised distillation, or distillation with unlabeled examples, see e.g. [2, 3]. )  It seems that, in order to apply the current approach, one requires the ground-truth labels and, thus,  one has to give up a big part of the power of knowledge distillation as a technique.)

To be clear, I still like the paper and I am leaning towards acceptance even if the scope of the paper is more limited, but I think it would be beneficial to the research community if the above comments were addressed.

[1] Understanding and Improving Knowledge Distillation [Tang∗, Shivanna, Zhao, Lin, Singh, H.Chi, Jain]
[2] Big self-supervised models are strong semi-supervised learners [Chen, Kornblith, Swersky, Norouzi, Hinton]
[3] Weighted Distillation with Unlabeled Examples [Iliopoulos, Kontonis, Baykal, Trinh, Menghani, Vee]

**Questions:**

— Does the proposed method and theory works well/applies in NLP datasets/binary classification contexts?
— Is there a way to apply this technique in the context of semi-supervised distillation?

---

> ### Author Response · Authors · 2023-11-17
> **Response to Reviewer AG5J**
>
> Thank you very much for your positive and valuable comments, which have helped us to improve the manuscript. We have implemented semi-supervised distillation, and demonstrated once again that the MCMI teacher outperforms the MLL teacher in the semi-supervised distillation as well. The respective results are included in appendix A.9 and shown in **“BLUE”** color in the revised version of the paper. Below, please find our point-by-point response to your comments.
>
>
>
> > Comment 1. The role of the teacher as a "provider of estimates for the unknown Bayes conditional probability distribution" is a theory for why distillation works that applies well mainly in the context of multi-class classification, and especially in the case where the input is images. (Indeed, there are other explanations for why knowledge distillation works, as it can be seen as a curriculum learning mechanism, a regularization mechanism etc see e.g. [1])
> In that sense, I feel that the author should either make the above more explicit in the text, i.e., explicitly restrict the scope of their claims to multi-classifcation and images, or provide evidence that their technique gives substantial improvements on binary classification tasks in NLP datasets (but even in vision datasets).
>
>
> **Response**: Yes, we agree that the experiments we have implemented so far are for image multi-classification problem. Having said this, our method, i.e., the MCMI estimator, is applicable to any DNN which outputs a probability distribution in scenarios where knowledge distillation (KD) is applicable, no matter how large or small the dimension of the probability distribution is. Note that the dimension of the probability distribution is 2 in binary classification and greater 2 in multi-classification. We have tested the MCMI estimator in the cases of 10 class classification (CIFAR 10), 100 class classification (CIFAR 100), and 1000 class classification (ImageNet). From our results (Table1 vs Table 2, and Table 9 vs Figures 14 to 19), it follows that in general, the accuracy performance gain offered by the MCMI estimator over the MLL estimator is even more when the number of classes is smaller. This can be explained by a larger percentage increase in the value of CMI when the number of classes is smaller. For example, the percentage increase in the value of CMI offered by the MCMI estimator in Table 1 is several magnitude larger than that in Table 2. Since CMI is well defined in the case of binary classification, there is no reason to believe that the MCMI estimator won’t work for binary classification problems to which KD is applicable. Of course, for those problems to which KD is applicable, how much gain the MCMI estimator would offer in comparison with the MLL estimator, especially for NLP datasets,  needs to be found out in future work.
>
> > Comment 2: One of the main reasons why knowledge distillation is such a popular technique, is because the teacher can generate pseudo-labels for new, unlabeled examples, increasing the size of the student's dataset. (This is known as semi-supervised distillation, or distillation with unlabeled examples, see e.g. [2, 3]. ) It seems that, in order to apply the current approach, one requires the ground-truth labels and, thus, one has to give up a big part of the power of knowledge distillation as a technique.)
>
> **Response:** No, that is not the case. The requirements for our approach are nothing more and nothing less than those for KD since we do not change the KD framework. The knowledge in terms of training samples and their labels required to train the teacher using MCMI is identical to that required to train the teacher using MLL. To further demonstrate this, We have implemented semi-supervised distillation, and demonstrated once again that the MCMI teacher outperforms the MLL teacher in the semi-supervised distillation as well. Please see appendix A.9 for details.
>
> > Comment 3: Does the proposed method and theory works well/applies in NLP datasets/binary classification contexts?
>
> **Response**: Please refer to our response to Comment 1.
>
> > Comment 4: Is there a way to apply this technique in the context of semi-supervised distillation?
>
> **Response**: Yes, the same as in the case of KD. Please refer to our response to Comment 2.

---

> > ### Comment · Reviewer_AG5J · 2023-11-19
> > **Response to Authors.**
> >
> > *"Since CMI is well defined in the case of binary classification, there is no reason to believe that the MCMI estimator won’t work for binary classification problems to which KD is applicable. Of course, for those problems to which KD is applicable, how much gain the MCMI estimator would offer in comparison with the MLL estimator, especially for NLP datasets, needs to be found out in future work."*
> >
> > I fully understand that the author's method is in principle applicable to binary classification and NLP datasets. However, in my experience, in the case of binary classification (and especially for NLP datasets) the improvements over vanilla distillation of almost all the distillation techniques I am aware of are particularly limited, if any. (Indeed, the limitations of distillation methods in binary classification settings is discussed in [4], where a method is provided that gives improvement in the cases where the input distribution has "subclasses"- structure.)
> >
> >
> > My larger point here is this: There are various reasons why distillation works depending on the setting. For multiclass-classification (and especially on vision tasks) it seems fairly plausible that  the "approximating the bayesian prior"-approach is the main source of the improvements. On the other hand, for binary classification tasks, it seems to me that the "curriculum learning / regularization"-nature of distillation is the main source of the improvement (that's why the improvements tend to be marginal). In that sense, I think it is very likely that your technique that aims to improve the Bayes conditional probability estimates might give no significant improvements in the latter setting — exactly because it is not optimizing for the "most valuable objective" for that setting.
> >
> > Regardless of whether my intuition above is correct or incorrect though, I think you should be a bit more careful with the claims you are making unless you are providing experimental evidence for it. In particular, unless you do experimentally show that you can get significant benefits over vanilla distillation in binary classification tasks (and especially NLP datasets), you should not assume your approach works in these settings.
> >
> > [4] Subclass Distillation [Rafael Muller, Simon Kornblith, Geoffrey Hinton]
> >
> >
> > *No, that is not the case. The requirements for our approach are nothing more and nothing less than those for KD since we do not change the KD framework. The knowledge in terms of training samples and their labels required to train the teacher using MCMI is identical to that required to train the teacher using MLL. To further demonstrate this, We have implemented semi-supervised distillation, and demonstrated once again that the MCMI teacher outperforms the MLL teacher in the semi-supervised distillation as well. Please see appendix A.9 for details.*
> >
> > Oh I see, thank you for the explanation and the extra experiments!
> >
> >
> > I raised my score to 8.

---

> ### Author Response · Authors · 2023-11-22
>
> We really thank the reviewer for reading our responses and raising our score.
>
> We have done some further experiments to address your concern about binary classification task as explained in the sequel.
> We acknowledge the fact raised by the reviewer in that “distillation works depending on the setting”; hence, the answer to the question that whether our method works for binary classification task might be elusive. As such, we have conducted some experiments on binary classification task to show the effectiveness of the MCMI teacher compared to the MLL teacher. Please kindly refer to **Appendix A.13** for the details. Briefly speaking, the MCMI teacher is also effective in binary classification task as well.
>
> We hope these extra experiments resolves all the concerns you have. Again, thank you for all your valuable comments.

---

### Official Review · Reviewer_Nadg · 2023-10-31

**Soundness:** 3 good
**Presentation:** 3 good
**Contribution:** 3 good
**Rating:** 6
**Confidence:** 3

**Summary:**

The paper proposes a method which aims to train the teacher to optimise the student. This is achieved through maximising the conditional mutual information between input and predicted label, conditioned on the true label. The approach demonstrates improved knowledge distillation on CIFAR100 and Imagenet using varies CNN architectures.

**Strengths:**

* The paper is very simple to understand and implement, which only a simple regulariser added to the training of the teacher model, which minimises the KL between the predicted probability and the average probability.
* The results are conclusive and well presented on ImageNet using plenty of architectures.
* The extension to few and single-shot experiments are nice.

**Weaknesses:**

In terms of weaknesses:
* I'm interested to read more about what the role of the CMI regulariser actually does, is it just decreasing the variance of the predictions? Or leading to a distribution with higher entropy? Does this method work just as well if you add an entropy regulariser?
* As far as I can tell, the value $T$ is not defined, is this for the softmax?

**Questions:**

* What is the value of $T$?
* Does the CMI loss just reduce the entropy?
* If so, is it possible that the same effect can be achieved by simply running this method with temperature scaling? I.e. drop the CMI term?
* With regards to 6.2. my understanding is that this is using the negative scores during training, so is this really zero-shot classification? Why do you expect this?
* Did you try varying different classes to drop?
* In Figure 3, why is the heat map on the terrier not on the body of the animal? Bottom, third from left.

---

> ### Author Response · Authors · 2023-11-18
> **Response to Reviewer Nadg (1/3)**
>
> Thank you for your time reading our paper, and your valuable comments. Please find your responses to your comments in the following.
>
> ## Weaknesses
> > Comment 1: I'm interested to read more about what the role of the CMI regulariser actually does, is it just decreasing the variance of the predictions? Or leading to a distribution with higher entropy? Does this method work just as well if you add an entropy regulariser?
>
> **Response:** Please refer to our reply to the comments of Reviewer oUDj for the role of the CMI regularizer and the reason why the MCMI estimator outperforms the MLL estimator. CMI is a completely different concept from the variance of the predictions and the entropy of the output distribution. Adding the entropy as a regularizer to the teacher’s loss does not improve the student’s accuracy, which was already demonstrated in the literature where it was shown that applying label smoothing and entropy penalty to the teacher’s training actually decreases the accuracy performance of the student in KD [1].
>
> Here, we further elucidate what is the **physical meaning** of CMI for DNNs. First, we note that the probability vectors $P_X$ for a specific class $y$ form a cluster in the output probability space of a DNN; we denote the centroid of this cluster by $Q^y$ (see equation (11) in the body of the paper).
>
> Now, referring to equation (11), by fixing the label $Y=y$, the conditional mutual information $I(X;\hat{Y}|Y=y)$ measures the average KL divergence between the centroid $Q^y$ and the output probability vectors $P_X$ for class $y$. This means that $I(X;\hat{Y}|Y=y)$ tells how all output probability vectors $P_X$ given $y$ are concentrated around its centroid $Q^y$. In this sense, a lower (higher) value for $I(X;\hat{Y}|Y=y)$ implies that the DNN’s predictions for class $y$ are more (less) concentrated around its centroid. On the other hand, as mentioned in equation (10), the CMI value for a classifier f denoted by CMI$(f)=I(X;\hat{Y}|Y)$ is related to $I(X;\hat{Y}|Y=y)$ as follows: $I(X;\hat{Y}|Y)=\sum_{y \in [C]} P_{Y}(y)I(X;\hat{Y}|Y=y)$. Hence,  $I(X;\hat{Y}|Y)$  tells how, on average, the DNN’s predictions are concentrated around the centroids of different classes. To better visualize this concept, please kindly see figure 6 (in the appendix).
>
> But what is our objective in this paper? As seen in equation (14), the proposed BCPD in the paper relies on **maximizing** the CMI value. Therefore, we are promoting the DNN’s predictions to be less concentrated (more dispersed) in each output probability’s cluster of one class, which can capture more contextual information from the input image compared to their MLL counterparts.
>
> On the other hand, variance and entropy are two different values with different meanings. Therefore, the value of CMI is unrelated to these quantities, and using another regularization method such as entropy regularization does not have the same effect as using CMI regularization.
>
> To further demonstrate this lack of relevance, **in appendix A.4.2**, we have compared teachers trained by three different regularization terms, namely CMI, entropy and label smoothing. The results further suggest that the CMI value is not related to entropy, and that for an effective KD, a high CMI value for the teacher matters, and not its entropy.
>
> > Comment 2. As far as I can tell, the value T is not defined, is this for the softmax?
>
> **Response:**
>
> Yes, $T$ is for softmax. Please note that we did not make any changes over KD framework; we only replace the conventional MLL teacher by the MCMI teacher. Therefore, all the notations employed in this paper, including $T$, are adopted from the knowledge distillation (KD) framework.
>
> [1]Müller, Rafael, Simon Kornblith, and Geoffrey E. Hinton. "When does label smoothing help?." Advances in neural information processing systems 32 (2019).

---

> ### Author Response · Authors · 2023-11-18
> **Response to Reviewer Nadg (2/3)**
>
> ## Response to Questions
>
> > Q1. What is the value of T?
>
> As discussed in the response to the second comment above, $T$ is indeed the temperature of the softmax function in KD. Additionally, the value of $T$ is the same as that used in the benchmark methods in their original paper. Hence, $T=1$ for all the KD methods tested over ImageNet dataset, and $T=4$ for the majority of the KD methods tested over CIFAR-100 methods.
>
> As also emphasized in the main body of the paper, one of the prominent advantage of the proposed method is “Plug-and-play” nature of MCMI teacher in that we do not change any hyper-parameters in the underlying knowledge transfer methods, all of which are the same as in the corresponding benchmark methods.
>
> > Q2. Does the CMI loss just reduce the entropy?
>
> Please refer to our response to comment 1.
>
> > Q3. If so, is it possible that the same effect can be achieved by simply running this method with temperature scaling? I.e. drop the CMI term?
>
> Thank you for your insightful comment. The temperature scaling (which is referred to as post-hoc regularization) has different effect compared to CMI regularization.
>
> In fact, the gain obtained via maximizing CMI is orthogonal to that obtained via tuning the $T$ values. More precisely, although by scaling $T$ for teacher, while fixing the $T$ for student, one can get some gain, this enhancement is independent of the gains achieved through maximizing the CMI value. To clarify, we have included **Appendix A.7**, where we conducted experiments with different $T$ values for the teacher and student, observing that the MCMI teacher consistently provides gains for any such combinations.
>
> > Q4. With regards to 6.2. my understanding is that this is using the negative scores during training, so is this really zero-shot classification? Why do you expect this?
>
> No, we are indeed aligned with the zero-shot setting for knowledge distillation as discussed in Section 3 of Hinton’s paper [1], where the authors specifically performed zero-shot distillation for the MNIST dataset. It's important to note that, unlike our method, the conventional knowledge distillation used in [1] does not work for other challenging datasets such as CIFAR-100 and CIFAR-10.
>
> We further note that our method is different from negative scoring method used in zero-shot learning which needs predefined semantic vectors for both seen and unseen classes and assigns positive scores to related semantics and negative scores to unrelated semantics during the evaluation stage [2]. We argue that our method is more generalized, as during the training process, teacher don’t know which classes will be omitted when distill knowledge to the student model.
>
> Lastly, we assume that you are asking why our teacher can lead to a student with better zero-shot performance. This is due to the fact that the information regarding the labels for the missing classes is still contained within $P_X ^*$ of the samples presented in the dataset. To elucidate, consider the following example in which the objective is to train a DNN to classify three classes: dog, cat, and car. Assume that the training samples for dog is entirely missing from the training dataset. Suppose that for a particular training sample from the cat class, $P_X ^*=[0.30,0.68,0.02]$, with the probability values corresponding to the dog, cat, and car classes, respectively. The value of 0.30 suggests that this sample exhibits some features resembling the dog class, such as a similarity in the shape of legs. Such information provided from the BCPD vectors for the samples of existing classes can collectively help the DNN to classify the samples for the missing class. We note that the one-hot vector for this sample is $[0,1,0]$, and therefore, such information does not exist when one-hot vectors are used as unbiased estimate of $P_X ^*$.
>
> [1] Hinton, Geoffrey and Vinyals, Oriol and Dean, Jeff ”Distilling the Knowledge in a Neural Network” stat. 2015
>
> [2] Huynh, Dat, and Ehsan Elhamifar. "Fine-grained generalized zero-shot learning via dense attribute-based attention." Proceedings of the IEEE/CVF conference on computer vision and pattern recognition. 2020.

---

> ### Author Response · Authors · 2023-11-18
> **Response to Reviewer Nadg (3/3)**
>
> > Q5. Did you try varying different classes to drop?
>
> Yes, please refer to Figure 14 to 19 in Appendix A.11, where you will find experiments on CIFAR-100 and CIFAR-10 datasets. Specifically for CIFAR-10, we conducted experiments with {1,2,…6} classes dropped from the dataset, each with two different combinations of the dropped classes. As observed, the gain obtained is independent of the classes dropped, and this gain is consistent for different numbers of dropped classes.
>
> > Q6. In Figure 3, why is the heat map on the terrier not on the body of the animal? Bottom, third from left.
>
> As discussed in our response to comment 1, the CMI is a class-based (cluster-based) concept, and cannot be evaluated by looking at a specific image. In other words, to evaluate the CMI value for a DNN, all the probability vectors (or the corresponding feature maps) within one class should be considered **as a whole**.
>
> If the CMI value for a given class is high, then the output probability vectors and their corresponding feature maps contain more information. These information show up in two formats: (i) the prototype, and (i) the contextual information which can be the background or the unique characters of the object in the images (noise of the cluster).
>
> Once looking at the Eigen-CAM for MLL teacher, for all the images the MLL teacher only captures the dog’s head (prototype); however, for the MCMI teacher this focus is more dispersed ranging from dog’s head, doge’s legs, and the background of the image (contextual information). As such, the MCMI teacher contains more information when we consider **all the images from one class**.
>
> Nevertheless, depicting only four images is not a good representation of the images in one class. To this end, in appendix A9 we sample and depict more images from four specific classes of the ImageNet training set to better illustrate this fact.

---

> ### Author Response · Authors · 2023-11-22
>
> Dear reviewer Nadg,
>
> We believe that our responses to your comments/questions along with the extra experiments we conducted fully address your concerns.
>
> Please let us know if you any further concerns; if not, we appreciate if you raise our score.

---

> > ### Comment · Reviewer_Nadg · 2023-11-22
> > **Response**
> >
> > Thank you for response and clarifying any misunderstanding, I've raised my score accordingly.

---

> > > ### Author Response · Authors · 2023-11-22
> > >
> > > We really thank the reviewer for reading our responses and raising our score.
> > >
> > > If you have any concerns or questions, please don't hesitate to reach out.

---

### Official Review · Reviewer_oUDj · 2023-11-01

**Soundness:** 3 good
**Presentation:** 3 good
**Contribution:** 2 fair
**Rating:** 6
**Confidence:** 3

**Summary:**

This work  builds upon the insights from the previous study on knowledge distillation [1], which implies that producing a good teacher model
similar to the optimal Bayes class probability $P^{*}_{X}$, is crucial for enhancing the performance of the student model. To convey this message, the authors propose a new training objective for the "teacher model" by introducing the empirical estimate of conditional mutual information as a regularizing term (MCMI).

The authors provide empirical evidence between MCMI and the accuracy of the student model; as the MCMI attains higher values, the the corresponding teach model obtains the highest accuracy. Furthermore, when using the teacher model trained with the MCMI regularizer, the corresponding teacher exhibits improved accuracy in most existing knowledge distillation algorithms. The proposed regularizer leads to improved performance of the student model in zero-shot and few-shot classification tasks  as well.

[1] A Statistical Perspective on Distillation - ICML 21

**Strengths:**

### Simple idea:

> In implementation sense, the idea looks simple and easy to implement this idea; introducing a estimate of the MCMI in Eq (2) is additionally necessary.

### Empirical improvement:
> It seems that the proposed objective for the teacher model can be integrated with existing knowledge distillation algorithms which mainly focus on the distillation objective in view of "student" model. The proposed regularizer for the 'teacher' model seems to be effective in enhancing the performance of the 'student' model trained with existing knowledge distillation algorithms.

**Weaknesses:**

### Less elaboration on relationship between conditional mutual information $I(X , \hat{Y} | Y)$ and optimal bayes classifier $P^{*}_{X}$

> While it is intuitively clear that using the conditional mutual information as the regularizer term can capture the contextual information of $X$ (Image) and provide additional information to a student model, the direct connection between conditional mutual information and the optimal Bayes classifier is less explained. I believe explaining this connection is important because this approach is motivated from the importance of optimal classifier $P^{*}_{X}$.

**Questions:**

* Q1.  Could you elaborately explain why minimizing $I(X , \hat{Y} | Y)$ can make the teacher model $f$ to be more similar to the optimal bayes classifier ?



* Q2. It seems that the proposed regularizer requires the pre-trained model as the teacher model and apply the further training to the teacher model with the proposed objective of Eq. (14). How do we set the number of iterations further training? Based on my understanding, since we expect this regularizer to make the teacher model contain additional information as well as to be properly certain (not overconfident), setting the number of iterations is important hyperparameters and might significantly affect the performance of student model.

---

> ### Author Response · Authors · 2023-11-17
> **Response to Reviewer oUDj (1/2)**
>
> We thank the reviewer for taking time to review our paper and provide valuable feedbacks. Below please find our responses to your comments.
>
> ## 1. Clarification
> > Comment 1:  This work builds upon the insights from the previous study on knowledge distillation [1], which implies that producing a good teacher model similar to the optimal Bayes class probability $P_X^*$, is crucial for enhancing the performance of the student. The authors propose a new training objective for the teacher by introducing the empirical estimate of conditional mutual information as a regularizing term (MCMI).
>
> **Response**: We appreciate the above comment and the work [1]. Indeed, the work [1] above along with other recent work in the literature cited in our paper helped us realize that training the teacher for the benefit of the student in the KD framework is a different task from training the teacher for its own accuracy performance. The former can be regarded as a process of estimating the unknown Bayes class probability $P_X^*$, and the teacher trained with the cross entropy objective function is actually a maximum log-likelyhood (MLL) estimate of $P_X^*$. The work [1] and others cited in the paper, however, do not suggest any new specific means to estimate $P_X^*$. Our paper introduces the concept of conditional mutual information (CMI) into the estimation of $P_X^*$ and propose, for the first time, an estimator called MCMI estimator which is different from and better than the MLL estimator.
>
> > Comment 2: The authors provide empirical evidence between MCMI and the accuracy of the student model; as the MCMI attains higher values, the the corresponding teach model obtains the highest accuracy. When using the teacher model trained with the MCMI regularizer, the corresponding teacher exhibits improved accuracy in most existing knowledge distillation algorithms. The proposed regularizer leads to improved performance of the student in zero-shot and few-shot tasks.
>
> **Response**: If we understand the above comment correctly, there is a slight misunderstanding here. Please refer to our response above for the difference between (1) estimating $P_X^*$ via training the teacher for the benefit of the student in the KD framework and (2) training the teacher for its own accuracy performance. The teacher trained with MCMI, acting as the MCMI estimator of $P_X^*$, increases the student’s accuracy across the board in all tested KD settings, with dramatic performance improvement in zero-shot and few-shot classification tasks, demonstrating that the MCMI estimator is better than the MLL estimator. However, as the CMI value increases, the teacher’s own accuracy would be degraded slightly. This, nonetheless, is not a problem since as mentioned above, estimating $P_X^*$ is a different task from training the teacher for its own accuracy performance, which is consistent with observations made in literature that teachers with higher accuracy on their own are not necessarily good teachers for the student in KD.
>
> > Comment 3: Less elaboration on relationship between conditional mutual information $I(X;\hat{Y}|Y)$ and optimal bayes classifier $P_X^*$
>
> **Response**: We truly appreciate this theoretic question in general. However, at this point, the formulation of this theoretic question is not clear at all for a couple of reasons: (1) the underlying joint distribution of (X,Y) and hence P_X^* are unknown, and any amenable assumption would go against the current philosophy of deep learning---if the joint distribution is known, there is no need for learning; and (2) CMI $I(X;\hat{Y}|Y)$ highly depends on the underlying deep neural network (DNN) which varies from one setting to another and for which deep understanding still lacks. Without certain assumptions on the underlying DNN, there might be no theoretical relationship between  $I(X;\hat{Y}|Y)$ and $P_X^*$. For example, when the underlying DNN is poorly designed---all output probability vectors huddle together,  $I(X;\hat{Y}|Y)$ could be always near 0 no matter what the joint distribution of $(X,Y)$ is. This setting is very different from the work [1] where the law of total probability for variance can be nicely applied in general. To overcome this difficulty, in this paper we instead choose the following approaches to justify and evaluate the proposed MCMI estimator:
>
> 1. Since the purpose of estimating $P_X^*$ is for the benefit of the student in KD, a good litmus test for an estimator is to see how the estimator can help the student improve its accuracy. In this regard, our proposed MCMI estimator is thoroughly compared to the MLL estimator and shown  superior to the MLL estimator across the board for all tested KD frameworks.
>
> 2. A synthetic Gaussian dataset is created and used to evaluate how close our MCMI estimate is to P_X^* . Please see appendix A.6 for details. Again, the results therein demonstrate the superiority of  the MCMI estimator to the MLL estimator in this synthetic setting as well.

---

> ### Author Response · Authors · 2023-11-17
> **Response to Reviewer oUDj (2/2)**
>
> ## 2. Response to Questions
>
> > Response to Q1.
>
> We guess you mean maximizing $I(X;\hat{Y}|Y)$. We won’t refer to the unknown Bayes class probability $P_X^*$ as the optimal Bayes classifier since the context for defining the optimality is not clear. In our view, the unknown Bayes class probability $P_X^*$ is defined by nature, or by human beings as a whole; it should be the average of all predictions made by a very large group of people. Intuitively, $P_X^*$ is influenced by different contextual information of the object in $X$ with respect to the corresponding image cluster, which would cause different people to make different predictions, and hence should stay a little away from the probability vector corresponding to the pure prototype (i.e., the object) which is the label one-hot vector in conventional deep learning, and the “centroid” vector $Q^Y$ defined in Eq. (11) in our case. It is the difference between $P_X^*$ and the prototype probability vector that reflects the variety of contextual information of the object. Maximizing $I(X;\hat{Y}|Y)$ to some degree would enable the teacher model f to extract more contextual information of the object into its output probability vector, thereby making it to be closer to $P_X^*$. This is indeed further confirmed by our results on synthetic Gaussian dataset in appendix A.6.
>
> > Response to Q2.
>
> For all the experiments we conducted in the paper, we fixed the number of fine-tuning epochs equal 20 for CIFAR-100 and 10 for ImageNet. This choice was made because we did not observe any gains in student’s accuracy beyond this number of epochs. However, it's important to note that the optimal number of epochs for fine-tuning may vary for different teacher-student pairs, and tuning this hyper-parameter could potentially yield additional improvements.

---

> ### Author Response · Authors · 2023-11-22
>
> Dear reviewer oUDj,
>
> We hope that our responses have fully addressed your concerns, if so we kindly ask the reviewer to raise our score; otherwise, we would like to resolve any concerns/ambiguity the reviewer has.

---

> ### Comment · Reviewer_oUDj · 2023-11-22
> **Response by Reviewer oUDj**
>
> Thanks for the author's response. I confirm all replies to my questions and have resolved my concerns. Thus, I raise my score from 5 to 6.

---

> > ### Author Response · Authors · 2023-11-22
> >
> > We sincerely appreciate the reviewer for reading our responses, and for raising our score.
> >
> > Thank you,
> >
> > The authors.

---

### Meta-Review · Area_Chair_2T3a · 2023-12-05

**Metareview:**

The paper proposes an improved teacher training method for knowledge distillation. The authors use a regularizer called maximum conditional mutual information (MCMI) which intutively forces the teacher predictive distribution to be more dispersed. The regularizer is motivated with an information-theoretic argument, and leads to improved student performance.

I would like to note that [this paper](https://arxiv.org/abs/2206.06661) also proposes a regularizer which improves teacher for KD. It is a different regularizer, but may ultimately help for related reasons, and should probably be discussed in the paper. [This paper](https://arxiv.org/abs/2106.05945) also provides relevant discussion on the effect of temperature in KD.

## Strengths

- The proposed regularizer is effective and simple / easy to implement
- The authors show positive empirical results
- I believe, the question of teacher training for student distillation is very important and under-explored

## Weaknesses

- I believe, it is still not completely clear why the proposed regularizer is the right way of making the teacher predictions closer to the correct Bayes classifier; the main argument is that it helps student performance empirically (according to [Response to Reviewer oUDj (1/2)](https://openreview.net/forum?id=yV6wwEbtkR&noteId=yOdFsov0vz)).
- Reviewer AG5J expressed concerns about the scope of the paper and applicability of the method

**Justification For Why Not Higher Score:**

While the paper is very interesting and the results are promising, the reviewers pointed out several weaknesses. In particular, it is not completely clear why the proposed regularizer is the correct thing to do for improving the teacher in KD.

**Justification For Why Not Lower Score:**

All reviewers are voting for acceptance. The paper is interesting, and makes a valuable contribution. I believe teacher training for knowledge distillation is a very important and under-explored topic.

---

### Decision · Program_Chairs · 2024-01-16

Accept (poster)